# ADAPTIVE GRAPH DENOISING WITH HARMONIC GROUPING

## ABSTRACT

Diffusion models have become the leading paradigm for generative modeling, yet their effectiveness in graph domains remains constrained by the instability of topological structures under noise. Existing noise schedulers, however, are typically fixed or heuristic, failing to adapt to either diversity across graph datasets or variability within a dataset. We formulate the challenge of graph diffusion scheduling from two perspectives: inter-diversity, referring to differences across graphs, and intra-variability, referring to variations within individual graphs. To address the challenge in a unified manner, we propose ADHaG (Adaptive graph Denoising with Harmonic Grouping), a novel scheduling framework that explicitly addresses these limitations. Each graph is assigned to a group defined by the Laplacian spectrum, within which we optimize a feature-conditioned schedule that parameterizes the base scheduler. Empirically, ADHaG demonstrates consistent improvements across both molecular and synthetic benchmarks. In particular, on the QM9 dataset, it attains dataset-level performance with respect to chemical validity; moreover, across all synthetic graph datasets, the orbit statistics consistently improve. These results demonstrate that adaptive scheduling can provide a principled approach to capture both inter- and intra-level variation of graphs.

## 1 INTRODUCTION

Diffusion-based generative models have emerged as the de facto paradigm for high-quality generation in continuous domains such as images, audio, and video, surpassing earlier approaches in both fidelity and diversity (Sohl-Dickstein et al., 2015; Ho et al., 2020; Nichol & Dhariwal, 2021). Beyond continuous data, diffusion models are increasingly applied to graph domains, showing promise in tasks such as molecular design (Alakhdar et al., 2024) and social network analysis (Yin et al., 2025; Xu & Ma, 2025), where the discrete and relational nature of graphs poses unique modeling challenges. Importantly, these challenges are compounded by the variability of graph data—both across domains, where molecular, planar, and social networks differ markedly in scale and structural motifs, and within a single dataset, where graph sizes and topological patterns can be highly diverse (You et al., 2018).

A distinctive property of diffusion models is that the forward process acts as both corruption and learning curriculum, determining how information is restored during reverse generation. The noise scheduler is central, governing training stability and sample quality. Nonlinear schedules such as cosine have been shown to better preserve semantic content, yielding higher fidelity than linear schedules (Nichol & Dhariwal, 2021). While noise scheduling has been extensively studied in continuous domains, analogous studies in graph diffusion remain limited.

A variety of approaches have been proposed to refine the forward process in graph diffusion: methods that directly transition node and edge states (Vignac et al., 2023), techniques that model node–edge interdependence (Liu et al., 2024), spectral strategies leveraging Laplacian eigenvalues and eigenvectors (Martinkus et al., 2022; Minello et al., 2024), and constraint-based approaches that enforce structural properties (Madeira et al., 2024; Su & Wu, 2025; Jo et al., 2024b). These studies underscore that forward process design is decisive for generative performance. Nevertheless, most existing methods rely on predefined schedules, failing to capture differences across graph domains and topological variability within datasets.

The limitation of predefined noise schedules motivates us to explicitly characterize the diversity in graph generation along two complementary perspectives. The first is *inter-diversity*: graphs from different domains—such as molecular, planar, and social networks—exhibit substantial variation in global–local structure, e.g., molecular ring systems and community modularity in social networks. The second is *intra-variability*: even within a single dataset, graphs differ significantly in node and edge counts as well as in their topological properties. Treating such heterogeneous structures under a single fixed schedule inevitably leads to under- or over-noising for particular graph types, thereby degrading both training efficiency and generative quality (Klepper, 2024; Yu & Zhan, 2025).

Based on this, we propose ADHaG (Adaptive graph Denoising with Harmonic Grouping), a novel scheduling framework that bridges the gap between noise scheduling and topological diversity, as shown in Fig. 1. ADHaG introduces a *harmonic grouping*, which is a spectrum-based grouping that leverages the Laplacian spectrum as a structural descriptor. In harmonic grouping, graphs are partitioned based on spectral similarities, exploiting the property that low-frequency components capture global structure while high-frequency components encode local details. Within each group, we estimate a learnable, feature-conditioned schedule scaling function $\gamma_g(\cdot)$ that is jointly optimized with the generative model in an end-to-end manner, allowing the forward process to align with both group-level spectra and per-instance geometry. This design establishes a paradigm distinct from prior approaches based on fixed schedules, reconstruction-centric spectral methods (Minello et al., 2024), or path-branching mechanisms (Jo et al., 2024b). Our main contributions are summarized as:

- Motivated by the challenges of inter-diversity and intra-variability in graph data, ADHaG introduces spectrum-aware noise scheduling that groups graphs by Laplacian eigenvalues, enabling principled, group-wise adaptation to structural diversity.

- ADHaG introduces feature-conditioned schedules scaling function $\gamma_g(\cdot)$ that parameterize a base scheduler (e.g., cosine). The schedule is trained jointly with the denoiser, allowing adaptation without changing the sampler.

- We validate ADHaG across molecular, synthetic, and social graphs. On QM9 (with explicit H), ADHaG achieved dataset-level performance, with validity, molecular stability, and atom stability that match or exceed the dataset statistics. On synthetic datasets, the orbit statistic (lower is better) is reduced by approximately 90% relative to the baseline. In most benchmarks, ADHaG consistently outperforms the previous baselines with fixed schedulers.

## 2 RELATED WORKS

### 2.1 GRAPH GENERATION WITH SPECTRAL-BASED DIFFUSION

Graph signal processing (GSP) is a mathematical framework that analyzes signals defined on graphs by adapting the classical Fourier Transform to the graph domain, enabling operations such as filtering, convolution, and downsampling across diverse domains with inherent irregularity and complexity (Shuman et al., 2013; Ortega et al., 2018; Dong et al., 2020). In general, low-frequency components corresponding to smaller eigenvalues have been more extensively utilized in graph representation learning, since they capture the dominant structural patterns that are crucial for most graph-level tasks while filtering out high-frequency noise (Wu et al., 2019).

The success of diffusion-based methodologies in generation tasks across diverse domains (Cao et al., 2024; Chen et al., 2024) has led to the active adoption of diffusion approaches (Sohl-Dickstein et al., 2015; Ho et al., 2020; Nichol & Dhariwal, 2021; Rombach et al., 2021) in the graph domain (Liu et al., 2023; Wang et al., 2025b). Research on score-based (Jo et al., 2022) or diffusion-based graph generation conditioned on graph spectrum has also emerged, such as SPECTRE (Martinkus et al., 2022), DiGress (Vignac et al., 2023), and others (Huang et al., 2023; Chen et al., 2023; Wu et al., 2023). DiGress (Vignac et al., 2023) introduced discrete denoising diffusion (Ho et al., 2020) strategies into Transformer architectures. Subsequently, various spectral-based graph diffusion methodologies (Luo et al., 2023; Cho et al., 2023; Shi et al., 2025; Wang et al., 2025a) have demonstrated promising results across diverse tasks including community graphs and molecular graphs. Recently, multi-conditional graph generation tasks have been successfully performed using Diffusion Transformer (Peebles & Xie, 2023) based architectures (Liu et al., 2024). Note that while various geometric graph diffusion studies utilizing 3D molecular structures have been recently published (Hooge-

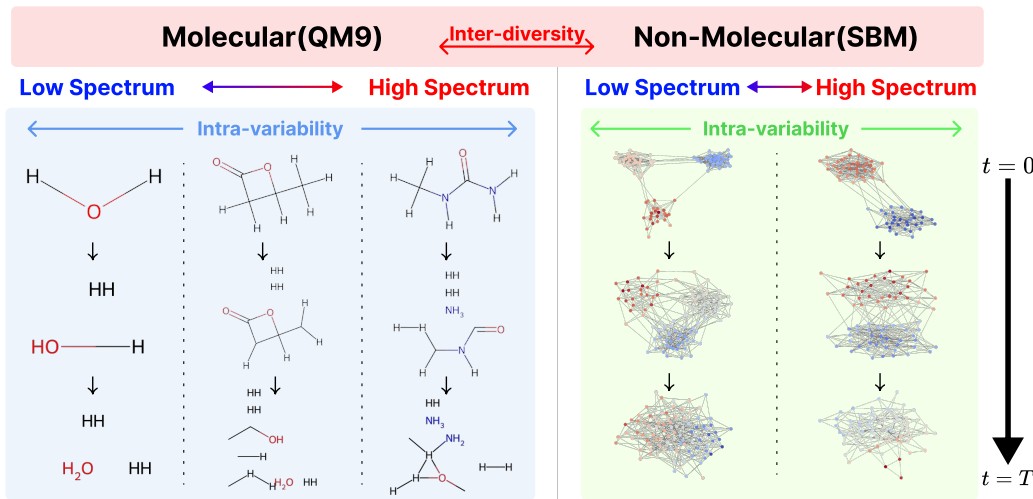

Figure 1: The visualization highlights the difficulties posed by distributional challenges in graph generation. Molecular (QM9) and non-molecular (SBM) datasets exhibit substantial inter diversity, while graphs within each dataset further present intra variability across spectral ranges. These observations underscore the need for methodologies capable of handling diverse and heterogeneous graph distributions.

boom et al., 2022; Xu et al., 2022; 2023; Jo et al., 2024a), the scope of this research does not address geometric information and focuses exclusively on 2D graphs without Euclidean distance information for molecular graphs. Further discussion on spectral GNNs and graph diffusion can be found in Appendix D.

## 2.2 Adaptive Schedulers in Diffusion-based Models

Among various attempts to enhance the performance and effectiveness of diffusion-based generative frameworks, research on improving diffusion noise scheduling has emerged (Hang et al., 2024; Guo et al., 2025). VDM (Kingma et al., 2021) can be considered an earlier study that enabled parametrized noise scheduling scales by utilizing monotonic neural networks, and Chen (2023) also studied the role of noise scheduling in diffusion research. However, most diffusion-based models have adopted fixed approaches, namely the linear scheduler of DDPM (Ho et al., 2020) or cosine scheduler of improved DDPM (Nichol & Dhariwal, 2021), which has been popular in recent days (Ho et al., 2022; Tevet et al., 2023; Lovelace et al., 2023). Recently, MuLAN (Sahoo et al., 2024) demonstrated that parametrization of multivariate noise schedules can directly improve both generation efficiency and training objectives. Nevertheless, to the best of our knowledge, no comprehensive research on noise scheduling that is adaptive to the graph spectrum has been conducted in graph structure-based domains.

## 3 Preliminary

In this section, we briefly overview two basics of our proposed method: (i) denoising diffusion (Ho et al., 2020) and (ii) graph Laplacian and its eigendecomposition for harmonic analysis (Shuman et al., 2013). For denoising diffusion, we outline the forward, noise scheduling, and backward processes, and for graph harmonic analysis, we summarize the connection between graph spectrum and its structural properties.

### 3.1 Diffusion Models

Diffusion-based generative models consist of a forward Markov chain that gradually adds noise to data and a parameterized reverse process that learns to remove noise. The forward process progressively transforms the original data $x_0$ into entirely noised $x_T$ over $T$ steps according to a predefined

noise schedule $\beta_t$. The (parameterized) reverse process predicts and removes noise at each step, reconstructing the original data distribution from $x_T \sim \mathcal{N}(0, \mathbf{I})$ back to $x_0$.

**Forward (noising).** For each step $t \in \{1, \ldots, T\}$, we define the signal preservation rate $\alpha_t = 1 - \beta_t$ and the cumulative signal rate $\bar{\alpha}_t = \prod_{s=1}^{t} \alpha_s$ with $\bar{\alpha}_0 = 1$. This allows us to express the forward process both conditionally and directly from the original data: $q(x_t|x_{t-1}) = \mathcal{N}(\sqrt{\alpha_t}x_{t-1}, (1 - \alpha_t)\mathbf{I})$ and $q(x_t|x_0) = \mathcal{N}(\sqrt{\bar{\alpha}_t}x_0, (1 - \bar{\alpha}_t)\mathbf{I})$, where $x_t$ gradually converges to a standard Gaussian distribution as $t$ increases.

**Noise scheduler.** The noise scheduler $\{\beta_t\}_{t=1}^{T}$ significantly affects the specific noising strategy of the forward process as well as training stability, sample quality, and convergence speed. For example, the cosine cumulative signal rate with normalization constant $\epsilon \geq 0$ is defined as

$$\bar{\alpha}_{base}(t) = \frac{\cos^2\left(\frac{\pi}{2} \cdot \frac{t/T+\epsilon}{1+\epsilon}\right)}{\cos^2\left(\frac{\pi}{2} \cdot \frac{\epsilon}{1+\epsilon}\right)} \in (0, 1] \tag{1}$$

**Reverse (denoising).** The reverse transition is parameterized as $p_\theta(x_{t-1}|x_t) = \mathcal{N}(\mu_\theta(x_t, t), \sigma_t^2 \mathbf{I})$. In the standard $\epsilon$-parameterization, the mean is given by $\mu_\theta(x_t, t) = \frac{1}{\sqrt{\alpha_t}}\left(x_t - \frac{1-\alpha_t}{\sqrt{1-\bar{\alpha}_t}}\epsilon_\theta(x_t, t)\right)$ with $\sigma_t^2 \in \{\beta_t, \tilde{\beta}_t\}$ and $\tilde{\beta}_t = \frac{1-\bar{\alpha}_{t-1}}{1-\bar{\alpha}_t}\beta_t$, where training is typically performed with uniform sampling of $t$ and $\epsilon_\theta$ regression loss. In the graph domain, discrete diffusion variants that use categorical noise (node/edge state transitions) instead of continuous Gaussian noise are also common (Vignac et al., 2023).

**Training objective.** The standard diffusion model is trained using different loss functions depending on the data type:

$$\mathcal{L}_{\text{MSE}} = \mathbb{E}_{x_0, \epsilon, t}\left[\|\epsilon - \epsilon_\theta(x_t, t)\|^2\right], \quad \mathcal{L}_{\text{CE}} = \mathbb{E}_{x_0, t}\left[-\log p_\theta(x_0|x_t)\right], \tag{2}$$

where the MSE loss is used for continuous data with Gaussian noise $\epsilon \sim \mathcal{N}(0, \mathbf{I})$ and $x_t = \sqrt{\bar{\alpha}_t}x_0 + \sqrt{1 - \bar{\alpha}_t}\epsilon$, while the cross-entropy (CE) loss is employed for discrete data (Austin et al., 2021).

## 3.2 GRAPH HARMONIC ANALYSIS

The spectrum of a graph (Chung, 1997; Shuman et al., 2013) describes global-local structures from a frequency perspective. Given an adjacency matrix $A \in \{0, 1\}^{n \times n}$ and degree matrix $D = \text{diag}(A\mathbf{1})$, we define the **normalized Laplacian** of a graph as $L = I - D^{-\frac{1}{2}}AD^{-\frac{1}{2}}$. From the eigendecomposition $LU = U\Lambda$, we obtain the *Laplacian eigenvalues* $\Lambda = \text{diag}(\lambda_1, \ldots, \lambda_n)$ with $0 = \lambda_1 \leq \cdots \leq \lambda_n \leq 2$.

Low frequencies capture global patterns while high frequencies capture local variations, with $\lambda_2$ (Fiedler eigenvalue) characterizing overall structural integrity.

This perspective enables quantifying structural heterogeneity across different datasets (e.g., molecular/planar/social) and diverse topologies within datasets (node/edge scales, community modularity, sparsity) through spectral statistics, serving as a key structural descriptor in graph learning and generation studies.

**Spectral graph diffusion.** In graph generation, spectral diffusion acts on graph eigenvalues in the frequency domain, in contrast to spatial diffusion acting on node features in the spatial domain. Given a graph with adjacency matrix $A$ and normalized Laplacian $L$, the spectral representation of a graph signal $x$ is $\hat{x} = U^T x$, where $\hat{x}$ contains the spectral coefficients. In general, the spectral diffusion process can be represented as $q(\hat{x}_t|\hat{x}_{t-1}) = \mathcal{N}(\sqrt{\alpha_t}\hat{x}_{t-1}, (1 - \alpha_t)\mathbf{I})$. The parameterized reverse process operates in the spectral domain: $p_\theta(\hat{x}_{t-1}|\hat{x}_t) = \mathcal{N}(\mu_\theta(\hat{x}_t, t, \Lambda), \sigma_t^2 \mathbf{I})$, where a neural network $\epsilon_\theta(\hat{x}_t, t, \Lambda)$ takes both the spectral coefficients and eigenvalues as input. The final reconstruction can be obtained by the inverse transform: $x = U\hat{x}$.

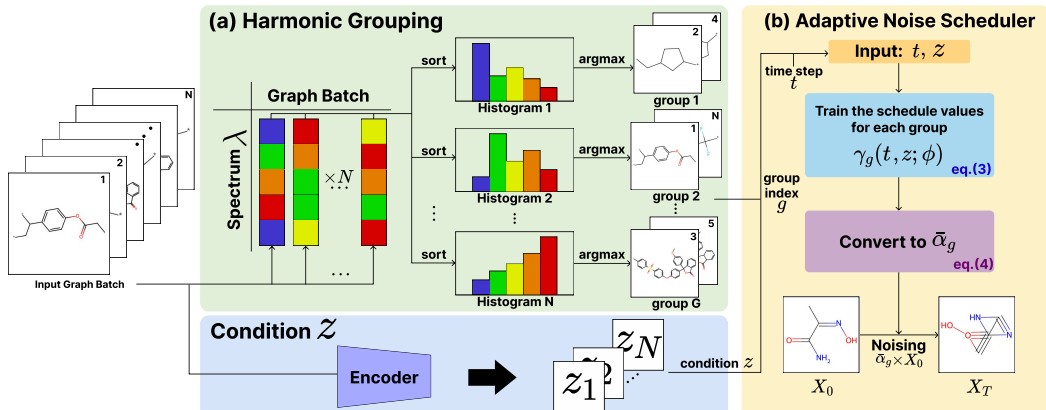

Figure 2: Overview of the proposed framework. **(a) Harmonic Grouping**. Given an input graph batch, we compute spectral eigenvalues and construct histograms to partition graphs into groups according to harmonic similarity (i.e., the closeness of dominant eigenvalues). **(b) Adaptive Noise Scheduler**. For each group $g$, the scheduler takes the time step $t$ and condition $z$ to train the adaptive noise schedule scaling function $\gamma_g(t, z; \phi)$, which is then converted to $\bar{\alpha}_g$ for applying the noising process during diffusion.

## 4 ADHaG

We propose ADHaG (Adaptive graph Denoising with Harmonic Grouping), which addresses the challenge that noise design for graphs is more subtle due to graph inter- and intra-level variation. ADHaG introduces two key components: (a) harmonic grouping and (b) adaptive noise scheduler with condition $z$. By leveraging spectral characteristics that encode global and local structural modes, our approach enables schedules that adapt to both inter-diversity and intra-variability. The schedules are optimized jointly with the reverse denoising process in an end-to-end manner.

### 4.1 HARMONIC GROUPING

To handle inter-diversity and intra-variability, we introduced a group partitioning in the graph frequency domain. First, we calculated all graph Laplacians and the top-$k$ largest eigenvalues in the whole dataset, and obtained the distribution of all eigenvalues. Next, we discretized the eigenvalue distributions into $G$ groups with equal width by dividing the range $[\lambda_{\min}, \lambda_{\max}]$ into $G$ equally-spaced intervals. Then, graphs were assigned to one of the groups according to the most dominant eigenvalue. Figure 3 shows the distribution of unnormalized eigenvalues from three datasets: Planar, 10K Polymers, and SBM used in the experiments. Note that the shapes of distributions are significantly different from each other, indicating the inter-diversity across graph domains.

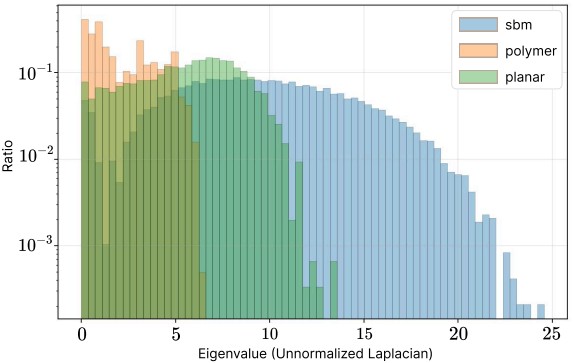

Figure 3: Eigenvalue distributions of SBM, 10K Polymers, and SBM datasets, showing distinct spectral characteristics across graph domains.

**Condition $z$.** To provide additional graph features for the adaptive noise schedulers, we utilized GNN-based encoder to embed the input graph $x$ into a conditioning vector $z$ (described in blue box of Figure 2). We set the dimension of $z$ to match the number of $G$, which is the number of partitioned groups. Based on the encoded $z$, the parameters in each scheduler of a group is defined and trained independently to better encode the intrinsic characteristics of each respective group.

## 4.2 Adaptive Noise Scheduler

Next, we introduce a novel scheduler to be adaptively parameterized depending on the graph spectrum. Specifically, we train $\gamma_g$ to be dependent on $z$ for the group index $g$ described in Section 4.1.

**Train each group schedule value $\gamma_g$.** For a base scheduler (e.g., cosine), we set a series of scheduling values $\{\log \bar{\alpha}_{\text{base}}[t]\}_{t=0}^T$ in ascending order with normalized time $0, \ldots, T$. We obtained $\log \bar{\alpha}_{\text{base}}(\tilde{t})$ at any intermediate time $\tilde{t} = t/T \in [0, 1)$ through linear interpolation. This initial scheduler serves as the base form, and through subsequent training, $\gamma$ adaptively adjusts the values at each time $t$.

We designed a scaling function $\gamma$, which is a scalar-valued function form $\gamma : [0, 1] \to \mathbb{R}_{\geq 0}$ for the base schedule $\bar{\alpha}$ be adaptively trained dependent on spectral groups $g$ and their latent representation $z$, aiming to be conditioned by spectral characteristics of graphs. Specifically, $\gamma_g$ with learnable parameter $\phi$ is defined as:

$$\gamma_g(\tilde{t}, z; \phi) = \gamma_{\min} + (\gamma_{\max} - \gamma_{\min}) \frac{\int_0^{\tilde{t}} \mathcal{F}_\phi(s, z) \, ds}{\int_0^1 \mathcal{F}_\phi(s, z) \, ds + \epsilon} \tag{3}$$

It maps the normalized time $\tilde{t}$ to a $\gamma$ value within the range $[\gamma_{\min}, \gamma_{\max}]$. The fraction computes a normalized cumulative ratio where $\mathcal{F}_\phi(s, z)$ acts as a learnable density function, $s$ is the integration variable, and small $\epsilon > 0$ for numerical stability.

**Convert to $\bar{\alpha}_g$.** Subsequently, $\gamma_g$ is used to scale and adjust the base scheduler $\bar{\alpha}_{base}(t)$, with $\tau$ serving as a *temperature* parameter to smooth the transitions.

$$\log \bar{\alpha}_g(\tilde{t}, z) = (\gamma_g(\tilde{t}, z; \phi)/\tau) \log \bar{\alpha}_{\text{base}}(\tilde{t}) \iff \bar{\alpha}_g(\tilde{t}, z) = \left(\bar{\alpha}_{\text{base}}(\tilde{t})\right)^{\gamma_g(\tilde{t}, z; \phi)/\tau} \tag{4}$$

For each group $g$, $\beta$ is defined based on the difference in $\bar{\alpha}_g$ between consecutive timesteps $t$ and $t - 1$ as follows:

$$\beta_g(t, z) = 1 - \bar{\alpha}_g(\tfrac{t}{T}, z)/\bar{\alpha}_g(\tfrac{t-1}{T}, z).$$

That is, the default form of the scheduler is set by the base scheduler of the original model, and during training, a set of $\{\gamma_g\}$ is learned to gradually update the base scheduler, with respect to each timestep $t$ and group $g$.

In contrast, our approach of starting with a conventional scheduler and updating it using scaling factor $\gamma$ for each timestep and group provides (a) training stability in early stages, (b) parameter efficiency, and (c) maximized performance by being customized to grouping and embedding conditions. Since conventional forms such as cosine schedulers has been known to be effective in graph domains, we argue that our methodology implements a principled approach by leveraging this prior knowledge.

Therefore, the forward and reverse processes with the adaptive scheduler scaling function $\gamma_g(\cdot)$ can be described as follows: $q(\hat{x}_t | \hat{x}_{t-1}) = \mathcal{N}(\sqrt{\bar{\alpha}_g(t, z)} \hat{x}_{t-1}, (1 - \bar{\alpha}_g(t, z))I)$ and $p_\theta(\hat{x}_{t-1} | \hat{x}_t) = \mathcal{N}(\mu_\theta(\hat{x}_t, t, z, g), \sigma_t^2 I)$.

## 4.3 Loss

As mentioned above, our objective function is based on $\mathcal{L}_{\text{base}}$ used in the original diffusion model and task, augmented with a weighted KL divergence loss to enable joint learning of our scheduler. Inspired by MuLAN (Sahoo et al., 2024), we introduced the weight term $w_{t,\phi}$ at each time step to amplify the loss when the difference between gamma values $\gamma_g(t, \cdot)$ and $\gamma_g(t - 1, \cdot)$ at consecutive timesteps becomes larger during training. Therefore, the final objective $\mathcal{L}_t$ incorporating both the denoising parameter $\theta$ and the scheduler parameter $\phi$ is formulated as follows,

$$\mathcal{L}_t(\theta, \phi) = \mathcal{L}_{\text{base}} + \sum_{t=1}^T w_{t,\phi} \mathcal{L}_{KL}(\theta; G) \tag{5}$$

Table 1: Multi-conditional generation of 10K Polymers based on Graph-DiT: Results on the synthetic score (Synth.) and three numerical properties (gas permeability for $O_2$, $N_2$, and $CO_2$). MAE is calculated between the input conditions and the properties of the generated polymers using Oracles. Best results per metric are in bold.

| Model | Validity ↑ (w/o rule checking) | Distribution Learning | | | | Condition Control | | | | |
|---|---|---|---|---|---|---|---|---|---|---|
| | | Coverage ↑ | Diversity ↑ | Similarity ↑ | Distance ↓ | Synth. ↓ | $O_2$Perm ↓ | $N_2$Perm ↓ | $CO_2$Perm ↓ | Avg. MAE ↓ |
| Graph GA | 1.0000 (N.A.) | 11/11 | 0.8828 | 0.9269 | 9.1882 | 1.3307 | 1.9840 | 2.2900 | 1.9489 | 1.8884 |
| MARS | 1.0000 (N.A.) | 11/11 | 0.8375 | 0.9283 | 7.5620 | 1.1658 | 1.5761 | 1.8327 | 1.6074 | 1.5455 |
| LSTM-HC | 0.9910 (N.A.) | 10/11 | 0.8918 | 0.7937 | 18.1562 | 1.4251 | 1.1003 | 1.2365 | 1.0772 | 1.2098 |
| JTVAE-BO | 1.0000 (N.A.) | 10/11 | 0.7366 | 0.7294 | 23.5990 | **1.0714** | 1.0781 | 1.2352 | 1.0978 | 1.1206 |
| DiGress | 0.9913 (0.2362) | 11/11 | 0.9099 | 0.2724 | 22.7237 | 2.9842 | 1.7163 | 2.0630 | 1.6738 | 2.1093 |
| DiGress v2 | 0.9812 (0.3057) | 11/11 | **0.9105** | 0.2771 | 21.7311 | 2.7507 | 1.7130 | 2.0632 | 1.6648 | 2.0479 |
| GDSS | 0.9205 (0.9076) | 9/11 | 0.7510 | 0.0000 | 34.2627 | 1.3701 | 1.0271 | 1.0820 | 1.0683 | 1.1369 |
| MOOD | 0.9866 (0.9205) | 11/11 | 0.8349 | 0.0227 | 39.3981 | 1.4019 | 1.4961 | 1.7603 | 1.4748 | 1.5333 |
| Graph-DiT | 0.8245 (0.8437) | 11/11 | 0.8712 | **0.9600** | **6.6443** | 1.2973 | 0.7440 | 0.8857 | 0.7550 | 0.9205 |
| + ADHaG (Ours) | 0.9844 (0.9522) | 11/11 | 0.8735 | 0.9363 | 7.2341 | 1.2872 | **0.6777** | **0.7836** | **0.6756** | **0.712** |

where $\mathcal{L}_{base}$ is either Mean Squared Error, Cross Entropy Error or other loss function, depending on the original loss term. The formulation of $w_{t,\phi}$ is described in Appendix C.

During inference, we construct $\bar{\alpha}_g$ using the selected group and embedding, and directly substitute it for the scheduler in the diffusion sampler. To ensure compatible utilization, we only modify the schedule without changing the sampler update formula.

## 5 EXPERIMENTS

### 5.1 TASKS

To evaluate the performance of ADHaG, we conduct experiments on two categories of graph generation benchmarks: synthetic graphs and (2D) molecular graphs. For synthetic graphs, we use the Community-Small (You et al., 2018), Planar (Martinkus et al., 2022), and SBM (Martinkus et al., 2022) datasets, which test the model's ability to capture structural regularity and community organization. For molecular graphs, we use the multi-conditional generation 10K Polymer dataset following GraphDiT (Liu et al., 2024) and the QM9 dataset, which is one of the standard benchmarks for molecule generation tasks, evaluating the model's capacity to generate chemically valid and diverse molecules. We compare against state-of-the-art graph diffusion models such as DiGress (Vignac et al., 2023) and GraphDiT by implementing ADHaG on these baseline models for a consistent comparison. Additional implementation details and hyperparameter settings are provided in Appendix A. We conducted three repeated experiments for all datasets except SBM and Community, and reported the average values.

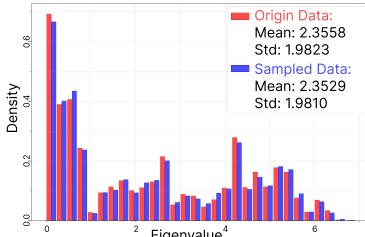

Figure 4: Eigenvalue distributions of original 10K polymers and generated samples, demonstrating that our model successfully reproduces the spectral characteristics with close agreement in mean and standard deviation.

### 5.2 MULTI CONDITIONAL MOLECULE GRAPH GENERATION

We trained our model based on Graph-DiT on the 10k Polymers dataset (in Table 1) and generated 10,000 samples for evaluation. Following Graph-DiT (Liu et al., 2024), we used Validity, Coverage, Diversity, Fragment-Similarity, FCD, and conditional control metrics (MAE and Accuracy). Validity shows the ratio after chemical validity filtering, with parenthetical values indicating unfiltered results. ADHaG achieved +16% validity improvement and 95% validity without rule checking. For multi-conditional polymer generation, ADHaG reduced average MAE by 0.2085 compared to the base Graph-DiT model across synthetic accessibility and gas permeability conditions.

Additionally, we verified whether the generated data exhibits a similar spectral distribution to the original dataset by comparing the eigenvalue distributions of generated samples and the original

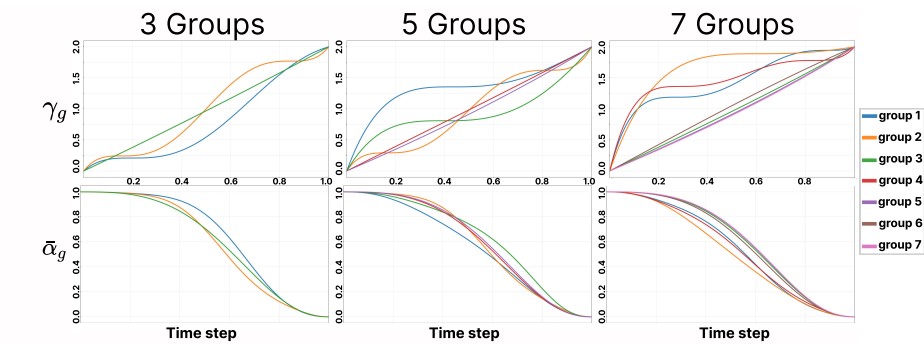

Figure 5: The trained noise schedules scaling factor $\gamma_g$ on the QM9 generation task. The plots show the evolution of $\gamma_g$ (top) and the corresponding $\bar{\alpha}_g$ (bottom) across time steps for different numbers of groups ($G=3, 5,$ or 7). As the number of groups increases, distinct scheduling patterns emerge, allowing more flexible modeling of various graph structures.

dataset (Figure 4). The strong correspondence of mean and standard deviation values between the generated samples and dataset eigenvalue distributions confirms that our model performs spectrum-aware generation effectively. A few examples of generated samples from each benchmark can be found in Appendix E.

## 5.3 MOLECULE GRAPH GENERATION

Table 2 summarizes the QM9 results. Despite using just 20% of DiGress's epochs, our method reduces NLL by 0.2. At the same epoch count, ADHaG reduced NLL by 3.6 and improved Validity by 0.4%. Increasing the number of groups raises Validity up to 99.7% but shows a relative decrease in NLL, suggesting that $G = 3$ is most suitable for this task. As shown in Figure 5, different groups exhibit distinct scheduling shapes, confirming that the scheduler adapts in a data-driven manner to the spectral characteristics of each group as intended, which directly contributes to the improved generation quality observed in our experimental results.

Table 2: Molecule generation on QM9. When applied to small graphs, with ADHaG, DiGress attains comparable results using only one-fifth of the training epochs and achieves improved performance under equal training epochs.

| Method | Epoch | NLL↓ | Valid ↑ | Unique ↑ |
|---|---|---|---|---|
| Dataset | – | – | 99.3 | 100 |
| SPECTRE | – | – | 87.3 | 35.7 |
| GDSS | – | – | 95.7 | 98.5 |
| DiGress | 1000 | 69.6 | 99.0 | 96.2 |
| + ADHaG ($G$=3) | 200 | 69.4 | 98.8 | 96.6 |
| + ADHaG ($G$=3) | 1000 | **66.0** | 99.4 | 96.3 |
| + ADHaG ($G$=5) | 1000 | 66.8 | **99.7** | 95.9 |
| + ADHaG ($G$=7) | 1000 | 66.9 | 99.3 | 95.9 |

**Ablation study of condition $z$** In this setting, we demonstrate the effects of condition $z$ and measure atom stability and molecule stability as defined in Hoogeboom et al. (2022), following DiGress evaluation protocol. Table 3 compares (i) without $z$, (ii) eigenvalue-based $z_{\text{eigs}}$ in Appendix B , and (iii) our proposed conditional embedding. The results show that

Table 3: Ablation study on QM9 generation according to $z$ conditioning with explicit hydrogens.

| Model | Valid ↑ | Unique ↑ | Atom stable ↑ | Mol stable ↑ |
|---|---|---|---|---|
| Dataset | 97.8 | 100.0 | 98.5 | 87.0 |
| DiGress (marginal) | 92.3 | 97.9 | 97.3 | 66.8 |
| + ADHaG w/o $z$ | 97.0 | 97.2 | 98.6 | 86.5 |
| + ADHaG $z_{eigs}$ | 96.6 | 97.3 | 98.5 | 85.7 |
| + ADHaG (Ours) | **97.5** | 96.7 | **98.9** | **89.4** |

**utilizing $z$ embedding consistently improves stability and overall metrics**, with our proposed method achieving performance comparable to the dataset level across all metrics except uniqueness.

## 5.4 GENERAL GRAPH GENERATION

Additionally, we evaluated general graph generation performance beyond molecular graphs using the DiGress-based model in Table 4. We used the same benchmarks as DiGress: Planar, SBM, and Community-small, with identical evaluation metrics including degree distribution (Deg.), clustering coefficient (Clus.), Laplacian spectrum (Spect.), and orbit/motif statistics (Orb.). Spectral metrics

Table 4: Comparison with other graph generative models using MMD metrics (the smaller, the better) on three synthetic datasets.

| | Community-Small | | | | Planar | | | | Stochastic Block Model (SBM) | | | |
|---|---|---|---|---|---|---|---|---|---|---|---|---|
| | Deg. ↓ | Clus. ↓ | Spect. ↓ | Orb. ↓ | Deg. ↓ | Clus. ↓ | Spect. ↓ | Orb. ↓ | Deg. ↓ | Clus. ↓ | Spect. ↓ | Orb. ↓ |
| GraphRNN | 0.0271 | 0.1072 | 0.0520 | 0.1469 | 0.0096 | 0.2985 | 0.0389 | 1.4022 | 0.0178 | 0.0151 | 0.0104 | 0.0351 |
| GRAN | **0.0013** | 0.0843 | 0.0282 | 0.0201 | 0.0202 | 0.2985 | 0.0248 | 0.1964 | 0.0135 | 0.0149 | 0.0034 | 0.0352 |
| GGSD | 0.0016 | 0.0590 | 0.0153 | 0.0142 | 0.0007 | 0.1881 | 0.0125 | 0.0047 | **0.0005** | **0.0115** | 0.0045 | 0.0289 |
| GSDM | 0.0099 | **0.0446** | **0.0131** | 0.0155 | 0.0220 | 0.0222 | 0.0096 | 0.0371 | 0.2295 | 0.2280 | 0.1578 | 0.2876 |
| GDSS | 0.0107 | 0.1060 | 0.0450 | 0.0356 | 0.0701 | 0.3025 | 0.0403 | 1.0345 | 0.2658 | 0.0442 | 0.0551 | 0.2780 |
| SPECTRE | 0.0079 | 0.1067 | 0.0460 | 0.0250 | 0.0008 | 0.0859 | 0.0147 | 0.0058 | 0.0044 | 0.0118 | **0.0015** | **0.0140** |
| DiGress | 0.0096 | 0.1035 | - | 0.0372 | **0.0005** | **0.0178** | **0.0020** | 0.0115 | 0.0166 | 0.0246 | 0.0064 | 0.1327 |
| +ADHaG (Ours) | 0.0170 | 0.0549 | - | **0.0042** | 0.0019 | 0.103 | 0.0052 | **0.0040** | 0.0261 | 0.0574 | 0.0097 | 0.0837 |

were not available for Community-Small in DiGress because its base implementation does not compute them. Planar was evaluated with $G = 7$ while the other two datasets used $G = 5$ in Appendix F.

When ADHaG was applied, we observed significant improvements compared to DiGress: Community-Small showed Clus. improvement from 0.1035 to 0.0549 and Orb. from 0.0372 to 0.0042; Planar achieved Orb. enhancement from 0.0115 to 0.0040; and SBM, demonstrated Orb. improvement from 0.1327 to 0.0837. The V&U&N index showed a 2.88-fold improvement over the baseline DiGress model, as shown in Table 5. All benchmarks showed meaningful orbit statistic improvements.

Overall, ADHaG consistently improves evaluation metrics across molecular, synthetic, and social graph domains. The trained group-wise schedules scaling function $\gamma_g(t, z; \phi)$ exhibit distinct profiles across datasets with different eigenvalue distributions, demonstrating adaptive behavior. The generated graphs show high accordance between spectral statistics and training data, suggesting improved structural faithfulness.

Table 5: Comparison with other graph generative models based on validity, uniqueness, and novelty metrics (the higher, the better) on SBM.

| Model | Stochastic Block Model (SBM) | | | |
|---|---|---|---|---|
| | Val.↑ | Uniq.↑ | Nov.↑ | V&U&N ↑ |
| GraphRNN | 0.13 | 1.00 | 1.00 | 0.13 |
| GRAN | 0.20 | 1.00 | 1.00 | 0.20 |
| GDSS | 0.01 | 1.00 | 1.00 | 0.01 |
| SPECTRE | 0.51 | 1.00 | 1.00 | **0.51** |
| GGSD | 0.49 | 1.00 | 1.00 | 0.49 |
| DiGress | 0.13 | 0.98 | 1.00 | 0.13 |
| +ADHaG | 0.38 | 1.00 | 1.00 | 0.38 |

## 6 DISCUSSION

**Contribution.** We present ADHaG, a spectrum-aware scheduling scheme that partitions graphs into harmonic groups via Laplacian and learns group-wise, feature-conditioned noise schedules directly from data. We formalize the forward-process noise scheduler as a spectrum-aware mechanism, thereby distinguishing it from conventional spectrum-agnostic predefined schedules. By embedding spectral information into the design of scheduling, ADHaG introduces an adaptive, data-driven approach that aligns the corruption process with intrinsic structural properties of graphs. This formulation establishes noise scheduling as a primary design dimension in graph generative modeling, rather than a fixed-shaped, highlighting its role in achieving both structural fidelity and generalization across domains.

Despite these advances, ADHaG has several limitations that suggest directions for future research. First, the method inherently depends on the choice of a base diffusion schedule, as ADHaG parameterizes a given cumulative SNR (e.g., cosine or polynomial). Consequently, the inductive bias of the selected base schedule remains influential, and developing base-free or more flexible parameterizations could further improve its generality. Another limitation lies in the need to predefine the number of harmonic groups $G$, which must be specified a priori. This fixed choice may limit adaptability across datasets with varying spectral characteristics. Designing data-adaptive strategies for selecting $G$ based on dataset size and spectral distribution represents a promising avenue for future work.

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

# APPENDIX

## CONTENTS

# A    EXPERIMENTAL SETTING

All graph Laplacians were used in the normalized form. We employed complete eigenvalues for all benchmarks. In cases where computing all eigenvalues becomes computationally challenging for sparse graphs with large number of nodes and low density (several cases observed in SBM dataset), we used the top-$k$ eigenvalues in ascending order of magnitude, where $k=32$. For embedding $z$ in the encoder, we utilized a 2-layer Graph Isomorphism Network (Xu et al., 2019) (other 2D graph neural networks can be used). The number of groups was set to 5 by default, with additional experiments conducted using 3 or 7 groups, and the number of different $\gamma$ values was set equal to the number of groups. The learning rate was set to $\frac{1}{10}$ of the denoising learning rate to prevent oscillation. We implement $\mathcal{F}_\phi(s, z)$ as a simple multilayer perceptron, though any integrable function can be used. All experiments were conducted using the official DiGress and Graph-DiT codebases, and both variants were trained on a single NVIDIA L40S GPU.

The official GitHub of base models (DiGress and Graph-DiT) are provided below.

Evaluation metrics followed the standard protocols of the given benchmarks. Specifically, for synthetic graphs including Community-Small (You et al., 2018), Planar (Martinkus et al., 2022), and SBM (Martinkus et al., 2022), we employed degree distribution (Deg.), clustering coefficient (Clus.), the occurrence frequency of all 4-node orbits (Orb.), and validity, uniqueness, and novelty (V.U.N). For molecule graphs, following existing methodologies, we conducted evaluation on 11 metrics belonging to Validity, Distribution learning, and Condition Control categories for 10K polymers (Thornton et al., 2012) following GraphDiT (Liu et al., 2024), and for QM9 (Ramakrishnan et al., 2014), we evaluated based on negative log-likelihood and V.U.N following previous studies (Martinkus et al., 2022; Vignac et al., 2023; Minello et al., 2025).

## A.1    SYNTHETIC GRAPHS

**Community-Small.** (You et al., 2018). The Community-Small dataset comprises 100 randomly generated community graphs, with each graph containing 12 to 20 nodes.

**Planar.** (Martinkus et al., 2022). This dataset comprises 200 planar graphs, each containing 64 nodes. These graphs are constructed through Delaunay triangulation applied to randomly and uniformly distributed point sets.

**Stochastic block model (SBM).** (Martinkus et al., 2022). This dataset features 200 Stochastic Block Model graphs with 2-5 randomly assigned communities, each containing 20-40 nodes. Within-community and between-community edge probabilities are 0.3 and 0.05, respectively.

## A.2    MOLECULE GRAPHS

**10K Polymers.**    For our first molecular generation benchmark, we adopt the multi-conditional molecule generation tasks from GraphDiT, consisting of numerical and categorical conditions. The benchmark encompasses polymer datasets for material design with gas permeability conditions ($O_2$, $CO_2$, $N_2$) with molecular properties and synthetic complexity scores. Following GraphDiT's experimental setup, we evaluate on molecules and polymers containing up to 50 nodes. For detailed dataset specifications, we refer readers to the original dataset (Thornton et al., 2012) and GraphDiT paper (Liu et al., 2024).

**QM9.**    As our second molecular generation benchmark, we employ the QM9 dataset (Ramakrishnan et al., 2014) following the standard evaluation protocol from previous works (Vignac et al., 2023; Minello et al., 2024). This dataset comprises 134k organic molecules containing up to 9 heavy atoms (C, O, N, and F). We adopt the established data split with 10k molecules each for validation and testing, and the remaining molecules for training.

---

GitHub: DiGress (official), Graph-DiT (official).

## B  WEIGHTED GROUP BAND AND $z_{eigs}$

The band mass is computed by weighting each eigenvalue with spectral power $p \geq 0$. Specifically, we set $w_i = \lambda_i^p$ for $p > 0$ or $w_i = 1$ for $p = 0$, and normalize the aggregated weights with a smoothing constant $\epsilon > 0$ to obtain the summary of the graph spectrum $\zeta \in \Delta^{G-1}$. Here, the eigenvalue range $[\lambda_{\min}, \lambda_{\max}]$ is uniformly partitioned into $G$ disjoint intervals $\{\mathcal{I}_1, \ldots, \mathcal{I}_G\}$ of equal width, so that each group corresponds to an equally spaced spectral band. Formally, the summary of the graph spectrum $\zeta$ is defined as:

$$\zeta_g = \frac{\sum_{i \in \mathcal{I}_g} w_i + \epsilon}{\sum_{g'=1}^{G} \left( \sum_{i \in \mathcal{I}_{g'}} w_i + \epsilon \right)}, \qquad w_i = \begin{cases} \lambda_i^p, & p > 0 \\ 1, & p = 0 \end{cases} \tag{6}$$

where the parameter $p$ controls sensitivity to high-frequency components (large $\lambda$). Setting $p = 0$ yields a simple count-based histogram, whereas $p > 0$ assigns greater mass to high-frequency bands, making the summary more discriminative for graphs with pronounced local fluctuations. The summary of the graph spectrum $\zeta$ is used as the conditioning $z_{eigs}$, encapsulating graph-spectrum information.

## C  TIME-DEPENDENT LEARNING OF $\gamma$

Since $\gamma_g$ is a *time-dependent* function, not all time steps are equally important during training. Following Sahoo et al. (2024), we multiply the core objective by a *temporal weighting* term $w_{t,\phi}$ derived from the difference of $\gamma$ between adjacent time steps, thereby allocating more learning signals to the *intervals with larger variations* in $\gamma$:

$$w_{t,\phi} = \frac{1}{2} T \Big( \exp\big(\gamma_g(\tfrac{t}{T}, z; \phi) - \gamma_g(\tfrac{t-1}{T}, z; \phi)\big) - 1 \Big). \tag{7}$$

This weighting helps the model align more rapidly in intervals where $\gamma$ changes sharply and critical information is reconstructed (e.g., in the later stages), while avoiding excessive intervention in the early stages so that the base structure is preserved. Finally, by multiplying the diffusion loss $\mathcal{L}_t(\theta, G)$ with $w_t$, the model *explicitly learns* the time-dependent structure of $\gamma$ (see $\mathcal{L}_t$ in Sec. 4.2).

## D  ADDITIONAL RELATED WORKS

### D.1  SPECTRAL GNNS

With the emergence of graph representation learning, frameworks combining GSP have been proposed to provide effective neural methods across various graph-structured domains (Defferrard et al., 2016; Kipf & Welling, 2017; Verma & Zhang, 2017; Wu et al., 2019). To address the fundamental limitation that computational cost increases with the number of nodes for graph Laplacians, Cheb-Net (Defferrard et al., 2016) and ChebNetII (He et al., 2022) utilized Chebyshev approximation to enable scalable application across diverse graph domains.

### D.2  GRAPH GENERATION WITH SPECTRAL-BASED DIFFUSION

As graph representation learning has developed in two major directions—spatial-based (or message passing (Gilmer et al., 2017)) and spectral-based approaches (Wu et al., 2020; Bo et al., 2023)—graph generation has similarly evolved with methods leveraging localized message propagation and graph spectrum (Guo & Zhao, 2022). Graph generation tasks were initially dominated by autoregressive methodologies such as GraphRNN (You et al., 2018), GRAN (Liao et al., 2019), CCGG (Ommi et al., 2022), and others (Kong et al., 2023), primarily focusing on community, citation, or biochemical network generation tasks. JT-VAE (Jin et al., 2018) and CGVAE (Liu et al., 2018) performed molecule graph generation based on variational autoencoders.

# E    GENERATED SAMPLES

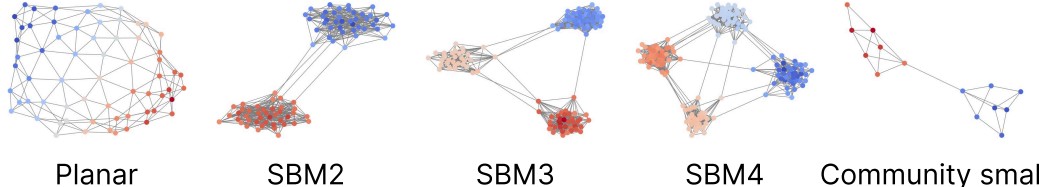

| Planar | SBM2 | SBM3 | SBM4 | Community small |

Figure 6: Graph samples generated by ADHaG. Each column corresponds to a different graph type: Planar, Stochastic Block Model (SBM) with 2 communities (SBM-2), SBM with 3 communities (SBM-3), SBM with 4 communities (SBM-4).

Figure 7: Molecular samples generated by ADHaG using the QM9 dataset. Each structure corresponds to a molecule sampled from the model, showcasing chemically valid and diverse functional groups.

To qualitatively evaluate the generative capability of ADHaG, we present sampled results from both graph and molecular domains. Figure 6 (top) shows examples of graphs generated by our model across different structures, including planar graphs, stochastic block models with varying community sizes (SBM-2, SBM-3, and SBM-4), interaction networks. The generated graphs capture the distinct topological characteristics of each type, such as modular community structure in SBMs.

In addition to graphs, ADHaG is also capable of generating valid molecular structures. As illustrated in Figure 7 (bottom), the generated molecules from the QM9 dataset exhibit chemically valid bonding patterns and diverse functional groups. These results demonstrate that ADHaG generalizes beyond abstract graph benchmarks and can be effectively applied to real-world structured data such as molecules.

# F   ABLATION STUDIES ON GROUP $G$ AND CONDITION $z$

Table 6: Comparison and ablation studies on the Planar dataset

| Model | Planar | | | |
|---|---|---|---|---|
| | Deg.↓ | Clus.↓ | Spect.↓ | Orb.↓ |
| DiGress | **0.0005** | **0.0178** | **0.0020** | 0.0115 |
| + ADHaG ($G$=3) | 0.0029 | 0.1218 | 0.0082 | 0.0169 |
| + ADHaG ($G$=5) w/o $z$ | 0.0022 | 0.0846 | 0.0059 | 0.0044 |
| + ADHaG ($G$=5) w/ $z_{eigs}$ | 0.0059 | 0.1447 | 0.0058 | **0.0034** |
| + ADHaG ($G$=5) | 0.0026 | 0.1044 | 0.0066 | 0.0043 |
| + ADHaG ($G$=7) | 0.0019 | 0.103 | 0.0052 | 0.004 |

Table 6 summarizes the Planar dataset results, showing that ADHaG improves multiple structural metrics over DiGress across various group sizes and condition $z_{eigs}$. While DiGress already achieves strong alignment with low-order structural statistics on the Planar dataset (i.e., degree, clustering, and spectral metrics), ADHaG yields notable improvements. In particular, the $G = 5$ with $z_{eigs}$ configuration substantially enhances the orbit metric, indicating more accurate recovery of higher-order structural patterns. Also, the $G = 7$ setting provides the most consistent performance across all reported metrics.

Table 7: Ablation of ADHaG on $O_2$–$N_2$–$CO_2$ conditional generation on 10K Polymers with Graph-DiT.

| Model | Valid ↑ | Unique ↑ | Novelty ↑ | Avg. MAE ↓ |
|---|---|---|---|---|
| Graph-DiT | 82.4 | **27.1** | 87.9 | 0.92 |
| + ADHaG $G$=1 | 91.7 | 25.8 | **91.4** | 1.03 |
| + ADHaG $G$=1 w/o $z$ | 98.1 | 5.6 | 55.1 | 0.73 |
| + ADHaG $G$=5 w/o $z$ | **98.5** | 18.5 | 78.5 | **0.82** |
| + ADHaG $G$=5 w/ $z$ | **98.4** | 21.5 | 89.6 | **0.71** |

Table 7 reports the ablation results of ADHaG under the $O_2$–$N_2$–$CO_2$ conditional setting in Graph-DiT. We vary the harmonic group count $G$ (with $G = 1$ meaning no grouping) and the use of the $z$ condition, and evaluate the models on Valid, Unique, Novelty, and Avg. MAE. In the $O_2$–$N_2$–$CO_2$ conditional generation setting, ADHaG—particularly the $G = 5$ configuration with the $z$ condition—achieves substantial improvements in both Valid and Avg. MAE, while largely preserving uniqueness and novelty. In contrast, the model without grouping and without the $z$ condition ($G$=1 w/o $z$) exhibits notable degradation in these metrics, indicating that ADHaG provides the most balanced performance relative to Graph-DiT.

Table 8: Ablation of grouping strategies on QM9 with explicit hydrogens.

| Method | NLL↓ | Valid ↑ | Unique ↑ |
|---|---|---|---|
| Dataset | – | 99.3 | 100 |
| DiGress | 69.6 | 99.0 | 96.2 |
| + Random Group | 69.8 | 98.8 | 96.6 |
| + Randequal Group | 69.9 | 99.2 | 96.4 |
| + Harmonic Group | **66.0** | **99.4** | 96.3 |

Table 8 reports the ablation results of different grouping strategies for molecular generation on QM9 with explicit hydrogens. Random Group assigns each graph to a random group, Randequal Group assigns graphs randomly while enforcing equal group sizes, and Harmonic Group uses the proposed Laplacian-based grouping. We evaluate the methods using NLL, Valid, and Unique metrics. The proposed harmonic (Laplacian-based) grouping achieves the lowest NLL and the highest validity

while fully preserving uniqueness, yielding overall superior performance compared to the DiGress baseline as well as the Random and Randequal grouping strategies.

# G EXTENDED EXPERIMENTS

Table 9: Comparison of molecular generation performance on the MOSES benchmark

| Model | Class | Val ↑ | Unique↑ | FCD↓ |
|---|---|---|---|---|
| VAE | SMILES | 97.7 | 99.8 | 0.57 |
| JT-VAE | Fragment | 100 | 100 | 1.00 |
| GraphINVENT | Autoreg. | 96.4 | 99.8 | 1.22 |
| ConGress | One-shot | 83.4 | 99.9 | 1.48 |
| DiGress | One-shot | 85.7 | 100 | 1.19 |
| ADHaG (ours) | One-shot | 86.6 | 100 | 0.84 |

Table 9 presents the molecular generation results on MOSES, comparing various graph genera-
tive models with the one-shot model equipped with the proposed ADHaG scheduler. The metrics
reported are validity (Val), uniqueness (Unique), and FCD. ADHaG demonstrates improved perfor-
mance over the one-shot baseline DiGress, particularly in FCD, while maintaining perfect unique-
ness. The proposed one-shot ADHaG model slightly increases validity and substantially reduces
FCD relative to DiGress, while maintaining 100% uniqueness, thereby achieving higher-quality
molecule generation even on large graph datasets.

Table 10: Comparison of molecular generation performance on the GuacaMol benchmark.

| Model | Class | Valid↑ | Unique↑ | KL div↑ | FCD↑ |
|---|---|---|---|---|---|
| LSTM | SMILES | 95.9 | 100 | 99.1 | 91.3 |
| NAGVAE | One-shot | 92.9 | 95.5 | 38.4 | 0.9 |
| MCTS | One-shot | 100 | 100 | 82.2 | 1.5 |
| ConGress | One-shot | 0.1 | 100 | 36.1 | 0.0 |
| DiGress | One-shot | 85.2 | 100 | 92.9 | 68.0 |
| ADHaG (ours) | One-shot | 88.8 | 100 | 93.0 | 68.4 |

Table 10 summarizes the molecular generation results on the GuacaMol benchmark, where all eval-
uations follow the protocol of DiGress. Incorporating the proposed ADHaG scheduler into the one-
shot DiGress model leads to improved validity and comparable or better FCD scores, while main-
taining perfect uniqueness. On the GuacaMol benchmark, DiGress equipped with ADHaG increases
validity from 85.2 to 88.8 while preserving 100% uniqueness, and also yields slight improvements
in both KL divergence and FCD scores, indicating an overall enhancement in molecular generation
quality.

Table 11: Extended results for molecular generation on the QM9 dataset.

| Model | Valid ↑ | Unique ↑ | Atom stable ↑ | Mol stable ↑ | Novel ↑ | FCD↓ | NSPDK↓ | KL↓ |
|---|---|---|---|---|---|---|---|---|
| Dataset | 97.8 | 100.0 | 98.5 | 87.0 | - | - | - | - |
| DiGress (marginal) | 92.3 | **97.9** | 97.3 | 66.8 | 33.4 | 11.6 | 0.28 | 0.23 |
| + ADHaG (Ours) | **97.5** | 96.7 | **98.9** | **89.4** | **34.0** | **10.7** | **0.27** | **0.22** |

Table 11 shows extended QM9 molecular generation results, additionally reporting Novel (Novelty),
FCD, NSPDK, and KL metrics. Except for FCD, NSPDK, and KL, all values are percentages. AD-
HaG is evaluated under the same protocol as DiGress. The model equipped with ADHaG achieves
dataset-level validity and atom/molecule stability, while reducing FCD, NSPDK, and KL scores. It
also maintains near-perfect uniqueness and slightly improves novelty, resulting in overall superior
molecular generation quality compared to DiGress.

## H  EIGENVALUE DISTRIBUTIONS

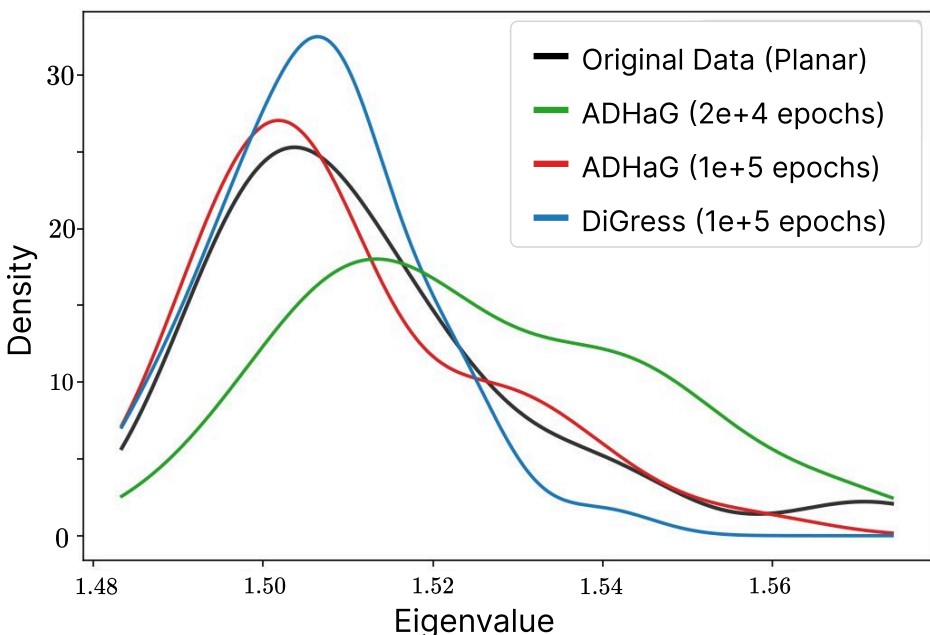

Figure 8: Eigenvalue distributions of original Planar and generated samples, demonstrating that our model successfully reproduces the spectruml characteristics with close agreement in mean and standard deviation.

Figure. 8 shows that ADHaG more faithfully recovers the eigenvalue distribution of the original Planar dataset, generating samples whose spectrum are closer to the ground-truth data than those produced by the baseline DiGress model. In addition, we compare the eigenvalue distributions of the original Planar graphs and those generated by ADHaG at different training epochs. With fewer training epochs (2e+4), the generated spectrum place relatively more mass on higher eigenvalue regions and deviate from the data distribution, whereas after sufficient training (1e+5 epochs) the mass shifts toward lower eigenvalues and the resulting distribution closely matches the original spectrum.

Table 12: Comparison with epochs using MMD metrics (the smaller, the better) on Planar datasets.

|  | Epochs | Deg. ↓ | Clus. ↓ | Spect. ↓ | Orb. ↓ | Val↑ | Uni↑ |
|---|---|---|---|---|---|---|---|
| DiGress | 100,000 | **0.0005** | **0.0178** | **0.0020** | 0.0115 | 0.75 | 1.0 |
| +ADHaG | 20,000 | 0.006 | 0.3112 | 0.0128 | 0.0265 | 0.35 | 1.0 |
| +ADHaG | 100,000 | 0.0019 | 0.103 | 0.0052 | **0.0040** | **0.8** | 0.96 |

Table 12 reports structural MMD metrics on the Planar dataset, comparing DiGress and DiGress with the proposed ADHaG scheduler. For generated planar samples from each model, we compute eigenvalue-based statistics and evaluate the degree (Deg.), clustering coefficient (Clus.), Laplacian spectrum (Spect.), and orbit (Orb.) distances, where lower values indicate better alignment with the reference distribution. On the Planar dataset, the ADHaG 100k-epoch model yields slightly higher structural MMDs (Deg./Clus./Spect.) than DiGress, but substantially improves the orbit MMD (Orb.).

# I  PSEUDO-CODE OF ADHAG

---

**Algorithm 1:** ADHaG Training

---

**1. Eigenvalue Computation**;
**Graph dataset:** $\mathbf{X} = \{X\}$;
**for** *X in* $\mathbf{X}$ **do**
  $\{\lambda\} \leftarrow$ normalize(eigendecomposition($X$));
  top-$k$ $\lambda$ values $\leftarrow$ sorted($\{\lambda\}$)$[: k]$;
  **for** $\lambda_k$ *in top-k* $\lambda$ *values* **do**
    $\lfloor$ $\{y_k\} \leftarrow$ calculate_density($\lambda_k$);

**2. Harmonic Grouping**;
Initialize $G$ groups by discretizing with equal-widths of whole range $[\lambda_{\min}, \lambda_{\max}]$;
**for** *X in* $\mathbf{X}$ **do**
  $\mathbf{g} = \arg\max_g \{y_k\}$;
  Groups$_{\mathbf{g}} \leftarrow$ assign($X$);

**3. Training and Sampling**;
**Noised sample:** $\mathbf{X}_t \leftarrow (X, \epsilon)$;
**Parameters:** $\theta$ for diffusion, $\phi$ for adaptive schedulers ($\gamma$ and GNN);
**Timesteps:** $t = 1, \ldots, T$;
**Diffusion:** $f_\theta(\mathbf{X}_t, t)$;
$G$ **schedulers** $\bar{\alpha}_{\text{base};g}$ and **any functions** $F_{\phi;g}$ for $g \in \{1, \ldots, G\}$;
**for** *each_step in training_steps* **do**
  // graph embedding
  $\mathbf{z} \leftarrow$ GNN$_\phi(\mathbf{X}_0)$;
  // adaptive scheduling
  $\gamma_g \leftarrow$ scaling($F_{\phi;g}(\cdot, \mathbf{z})$) ;                        // (Eq. 3)
  // schedulers
  $\log \bar{\alpha}(t) \leftarrow \gamma_g \cdot \log \bar{\alpha}_{\text{base}}(t)$ ;      // (Eq. 4)
  // sampling (forward process)
  $\mathbf{X}_t \leftarrow$ sample($\log \bar{\alpha}_t, \mathbf{X}_0$);
  // denoising (reverse process)
  $\mathbf{X}_{t-1} \leftarrow f_\theta(\mathbf{X}_t, t)$;
  // loss
  **loss** $\leftarrow$ DiffusionLoss($\epsilon_t, \hat{\epsilon}_t$) + weighted_KL_loss($\mathbf{X}_t, \hat{\mathbf{X}}_t$) ;      // (Eq. 5)
  update $\theta, \phi$;

---

## J   COMPARE WITH BASE MODEL

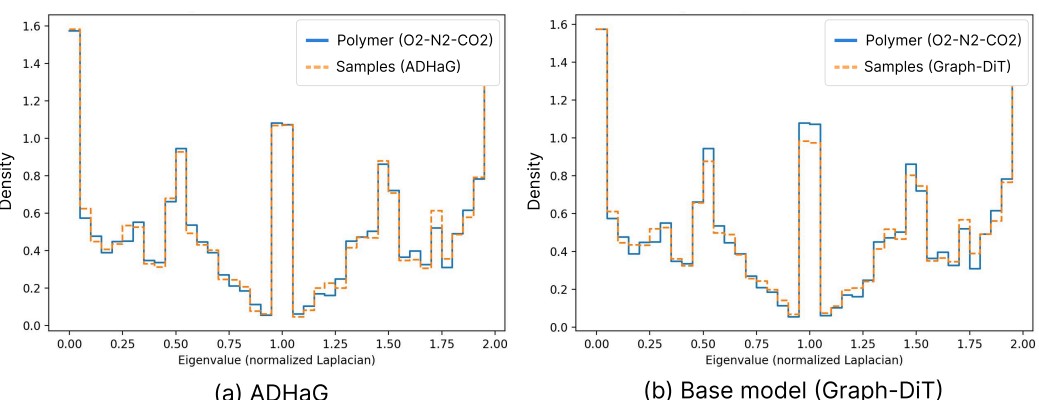

(a) ADHaG                                    (b) Base model (Graph-DiT)

Figure 9: **Eigenvalue distribution histogram of generated samples.**

As shown in Figure 9, the eigenvalue distributions of the generated graphs under (a) ADHaG (left) closely match those of the ground-truth polymer dataset, while the (b) base Graph-DiT model (right) exhibits visibly larger deviations, particularly in several spectral regions. This comparison shows that the adaptive, group-wise scheduler of ADHaG helps preserve the spectral structure of the data more faithfully than the base model.

Table 13: Comparison with graph generative performances using 1D Wasserstein distance (W.D) metrics on two molecule datasets.

|  | Qm9 with H | Polymer |
| --- | --- | --- |
|  | W.D ↓ | W.D ↓ |
| Basemodel(Graph-DiT) | 0.003281 | 0.005287 |
| ADHaG (Ours) | **0.002813** | **0.004056** |

Table 13 quantifies this effect using the 1D Wasserstein distance between the eigenvalue histograms of generated and test graphs. Across both QM9 with explicit hydrogens (QM9 with H) and Polymer, ADHaG consistently achieves lower distances than the base Graph-DiT model, indicating more accurate spectral alignment and improved structural consistency in the generated graphs.

## K  ARCHITECTURE DETAILS

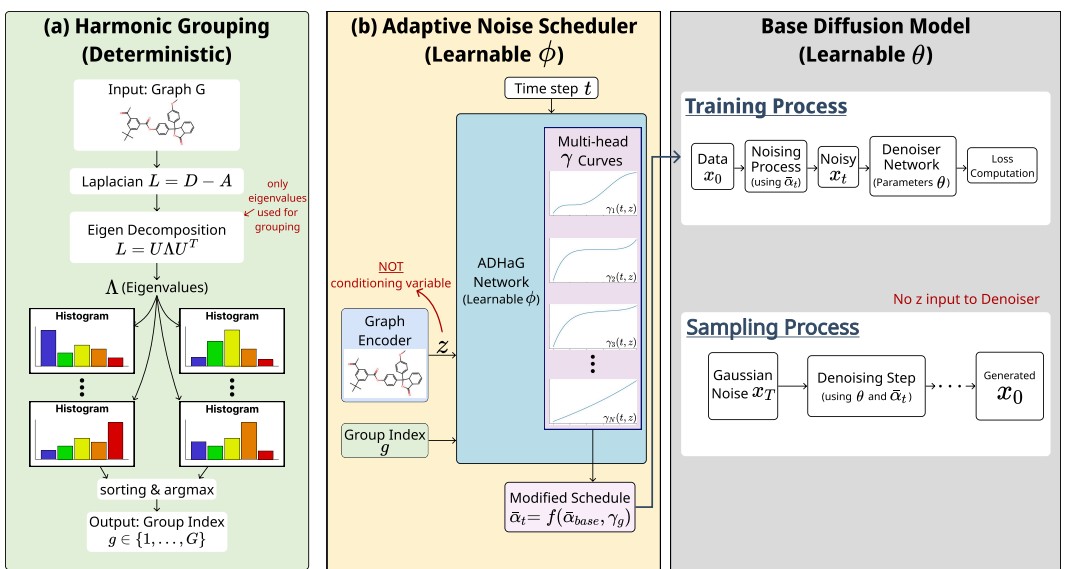

Figure 10: **Detailed illustration of the proposed framework.**

Figure 10 provides a detailed version of the main framework illustration. It expands Figure 2 by explicitly showing how eigenvalues are computed to deterministically assign each graph to a group, how the ADHaG network generates group-specific scheduling curves, and how the resulting schedules are applied to the underlying diffusion model without altering its original architecture.

## L    LLM USAGE

During the manuscript preparation, we used OpenAI's GPT5 (https://chatgpt.com/) and Claude's Sonnet4 (https://claude.ai/), Large Language Models, to proofread our work. Our interaction with the LLM was iterative and focused exclusively on improving the quality of the writing. We affirm that the LLM served as an assistive tool and did not contribute to core research ideas, experimental design, analysis, and results presented in this paper. The final scientific content and all claims made in this paper are the sole responsibility of the authors.

