# OpenReview forum: "Adaptive Graph Denoising with Harmonic Grouping"
_ICLR.cc/2026/Conference — Submitted to ICLR 2026_

### Official Review · Reviewer_Uz6U · 2025-10-30

**Soundness:** 2
**Presentation:** 1
**Contribution:** 1
**Rating:** 2
**Confidence:** 4

**Summary:**

The paper proposes a new data-dependent and adaptive noise scheduling scheme for graph diffusion models, such that groups of graphs with similar structural characteristics and properties have their own noise schedules that are learned with a neural network. The grouping is done using the normalised Graph Laplacian eigenvalue spectrum, grouping similar spectrum graphs together, into G distinct groups. Then, an adaptive noise scheduler is designed that takes as input the group g as well as a vector z that describes some features of the particular graph, and can be obtained through a GNN encoder. In practice, the adaptive schedule is implemented as a modulation of a standard cosine schedule. The loss function is also modified to take into account changes in the noise level value at a particular diffusion step $t$, such that changes in the noise level also amplify the loss weighting. The experimental validation consists of synthetic graphs and molecular graphs with the 10k Polymer dataset and QM9. The method achieves improvements over various baselines.

**Strengths:**

- The method seems to achieve valid improvements over many of the baselines in many of the metrics.
- The idea of adjusting the noise schedule for different types of graphs or graph datasets makes sense, and as such the method is quite well motivated
- The way the adaptive noise scheduler itself is implemented makes sense and seems like an approach that can bring benefits. I think that the research direction and the intuition about the problem statement makes sense.

**Weaknesses:**

The paper also does have some weaknesses:
- Overall, the paper was quite difficult to follow, and some details of the method are not very explicitly explained, and are potentially inconsistent with different parts of the paper. See the questions for more.
- The usage of the z vector is particularly confusing to me. As I understand it, it is a GNN embedding of the graph. So we condition the schedule on individual graphs at training time. How do we get the $z$ at inference time? Further, how do we choose $g$ at inference time?
- Further, it seems that the $g$ and $z$ conditioning are also given as input to the denoiser itself. This raises the question: Do we need them as input to the noise scheduling function $\gamma$? It is not really at all obvious why would they provide added benefit over just conditioning in the denoiser, raising the need for (at least) the following ablations to make sure that both $g$ and $z$ are useful even for the base denoiser, as well as for the addition of the $\gamma$:
1) $\gamma=1$ and $g$ conditioning for the denoiser but no $z$ conditioning
2) $\gamma=1$ and $z$ conditioning for the denoiser but no $g$ conditioning
3) $\gamma=1$, and both $z$ and $g$ conditioning for the denoiser
4) $\gamma$ conditioned on $g$, and both $z$ and $g$ conditioning for the denoiser
5) $\gamma$ conditioned on $z$, and both $z$ and $g$ conditioning for the denoiser
6) $\gamma$ conditioned on $g$ and $z$, and both $z$ and $g$ conditioning for the denoiser
- There is quite a bit of added complexity, and for the metrics in which the metrics outperforms the baselines, it is not obvious why this would be the case. Ideally, the paper would have a hypothesis for what kinds of issues does the graph diversity problem cause in the baseline models, and analyze quantitatively how the proposed method overcomes this issue and beats the baselines. Otherwise, looking at it from the outside, it is seems probable that the improvements on some measured variables could be due to small details like architecture design, hyperparameter tuning, including the $z$ conditioning to the denoiser, etc. This makes it difficult to have any confident takeaways from the paper.

**Questions:**

- It was a bit unclear to me while reading the paper how does the graph partitioning actually work. Do I understand correctly that we 1) calculate the normalized graph Laplacian eigenvalues for each graph separately, chunk the range of possible eigenvalues into G groups, and classify a graph in a particular cluster based on which group does it have the most eigenvalues? I am not sure if this was clearly stated, but Figure 2 seems to imply something like this.
- What are we seeing in Figure 3? All the eigenvalues of all the graphs in a dataset? How come we have negative values and values over 2 in the plot, given that the normalized graph Laplacian eigenvalues are in the range [0,2] (to the best of my understanding).
- What exactly is the role of the z vector? Why is it needed as an input to the schedule?
- "for graphs, the diffusion process typically operates on the spectral domain. " -> what does this mean, since the paper seems to actually use the Digress formulation of graph generation?
- Do the authors have an intuition for why do the schedules for the different groups end up like they do in Figure 5? The curves seem quite random to me, but perhaps there is some structure that is interesting?

Due the reasons raised in the weaknesses, I am starting with a reject. I think that the paper would benefit from clearer writing and explanation of the method, more focused experiments showing the benefits, and potentially streamlining the method to the minimum added complexity that is necessary to achieve the potential improvements from the adaptive schedule.

---

> ### Author Response · Authors · 2025-11-22
>
> We appreciate your thoughtful comments and meaningful suggestions for improving our manuscript. We have carefully addressed your key concerns below.
>
> **[W1, Q4] Legibility issue**
>
> Thank you for the feedback. We have revised the manuscript to clarify several methodological details and to ensure consistency across sections. In particular, we refined the text in Sections 4.1 (methodology of harmonic grouping) and 5.1 (experimental tasks) and corrected minor typographical and redundant phrases; the updated passages are highlighted in red. Regarding the specific issue raised at line 210 (Section 3.2), we modified the sentence from “for graphs, the diffusion process typically operates on the spectral domain.” to the more precise “In graph generation, spectral diffusion acts on graph eigenvalues in the frequency domain, in contrast to spatial diffusion acting on node features in the spatial domain.” Although our spectral formulation is inspired by DiGress, it is consistent with the general treatment of graph diffusion in the frequency domain used in models such as GDSS and SPECTRE. We also added a pseudocode of ADHaG training procedure in Appendix I. If there are additional parts that would benefit from further clarification, we will gladly clarify any additional points.
>
> **[W2, Q3] Clarification of the role of $z$**
>
> We appreciate the reviewer’s request for clarification. The vector $z$ serves as a conditioning feature obtained by encoding the input graph $X$ with a shallow GIN encoder; its role is to provide localized graph information when learning the adaptive noise scaler $\gamma$. This encoder can be replaced with other GNN, and we have clarified  this point in Section 4.1 and added pseudocode in the Appendix I. Importantly, $z$ is used only during training of $\gamma$. After finishing $\gamma$ training, $\gamma$ is fixed and does not depend on $z$ during inference anymore. Similarly, the group index $g$ is a predefined hyperparameter, and it remains unchanged during both the training and inference processes."
>
> **[W3] The form and contribution of $g$ and $z$, with their ablation studies**
>
> **Table: Ablation study on multi conditional generation according to $z$ conditioning and Harmonic grouping.**
>
> | Model | Valid ↑ | Unique ↑ | Novelty ↑ | Avg. MAE ↓ |
> |-------|---------|----------|-----------|------------|
> | Graph-DiT | 82.4 | **27.1** | 87.9 | 0.92 |
> | | | | | |
> | + ADHaG $G=1$ | 91.7 | 25.8 | **91.4** | 1.03 |
> | + ADHaG $G=1$ w/o $z$ | 98.1 | 5.6 | 55.1 | 0.73 |
> | + ADHaG $G=5$ w/o $z$ | **98.5** | 18.5 | 78.5 | **0.82** |
> | + ADHaG $G=5$ w $z$ | **98.4** | 21.5 | 89.6 | **0.71** |
>
> We thank you for your valuable comments and for suggesting ablation experiments that benefit our research. First, we would like to clarify the roles and implementation of $g$ and $z$. The group index $g$ is generated according to the predetermined number of groups and allows the scheduler scalar $\gamma$ to be identified for each $g$ during the denoising process. Therefore, we believe it is difficult to attribute the performance improvements in this study solely to the conditioning of the denoiser. We have added pseudocode in Appendix I to more clearly describe the form and roles of $g$ and $z$ in the overview shown in Figure 2.
>
> While we agree that your suggestion is highly valuable and that additional experiments to clearly demonstrate the roles of $g$ and $z$ are necessary, it is difficult to directly implement your proposal given our methodology described above. Since $z$ is the basic required representation of the adaptive scalar gamma, we cannot apply $z$ conditioning when $\gamma$=1 (i.e., when $\gamma$ learning is not performed). Furthermore, when we understand $g$ as grouping, in settings without grouping, $G$ (the number of groups) equals 1, and $G=0$ is not possible. Instead, considering your concerns, we conducted ablation experiments on the presence/absence of grouping and $z$ conditioning in the ADHaG scheduler. The detailed experimental settings are as follows:
>
> We conducted experiments based on multi-conditional (O2-N2-CO2) graph generation on the Polymer dataset. First, corresponding to #2 and #5, we experimented with no grouping (G=1) with $z$ conditioning. Second, we experimented with $G=1$ w/o $z$, where both grouping and $z$ conditioning were removed. Third, corresponding to #1 and #4, we experimented with $G=5$ without $z$ conditioning. Finally, corresponding to #3 and #6, we experimented with both grouping and $z$ conditioning present. These results are shown in the above Table, and we added it to Appendix F as Table 7.
>
> We would appreciate your understanding that we conducted the above experiments with maximum consideration for the intent and purpose you suggested. Please let us know if you think additional experiments are needed.

---

> ### Author Response · Authors · 2025-11-22
>
> **[Q1] Clarification of the way of graph partitioning**
>
> Yes, your interpretation is consistent with our intended procedure. To remove any ambiguity in the original description, we have revised the explanation in Section 4.1 (harmonic grouping) to present the partitioning procedure more explicitly. Additionally, we have included pseudocode in Appendix I for further clarification.
>
> **[Q2] Intuition behind Figure 3**
>
> The intention behind Figure 3 is to illustrate the inter-diversity and intra-variability highlighted in the Introduction by showing that different datasets exhibit markedly different eigenvalue distributions. Since the scaling used for the $x$ and $y$ axes in the current Figure 3 does not seem to clearly convey our intention, we modified the metric representation of the $x$-axis and $y$-axis. We changed the $x$-axis to the distribution of unnormalized raw eigenvalues, and the $y$-axis to the ratio representing the proportion in each dataset. We also changed the plot type to a histogram, which is more suitable for our original intention. Given that the raw eigenvalue range varies significantly across datasets, we can observe that the eigenvalue range differs for each group even when using the same $G$ value.
>
> **[Q5] The interpretation of Figure 5**
>
> As you pointed out, the shapes of the learned schedulers in each eigenvalue group exhibit diverse patterns. We have formulated one hypothesis regarding the cause of these patterns. From Figures 3, 4, and 8, we observe that most graphs concentrate in the low–eigenvalue region, while the high–eigenvalue region forms a relatively sparse tail. With a single global scheduler, the training signal is dominated by this dense low–eigenvalue head, and graphs in the high–eigenvalue tail behave like outliers or unseen points: they are present in the dataset, but they have much less influence on the diffusion and denoising dynamics, so the model does not accurately capture that part of the spectrum. Figures 3 and 8 support this interpretation. The original data already places most of its mass on low eigenvalues, and a conventional one-scheduler model tends to focus even more heavily on that region, under-representing the high–eigenvalue tail in the generated samples.
>
> In other words, the intra-diversity along the spectrum is large, but a single schedule is asked to cover all spectral regimes at once. ADHaG addresses this issue by partitioning the spectrum into several groups and learning a separate scheduler for each eigenvalue band. Once the spectral regions that were handled by a single global scheduler are split across groups, each scheduler of groups can specialize to the data density of its own group. This leads to more accurate handling of outlier-like high–eigenvalue graphs and, empirically, to improved overall validity compared to the base models. We emphasize that this remains a hypothesis based on empirical evidence, and we have added Figure 8 in the appendix to make this intuition explicit.

---

> > ### Comment · Reviewer_Uz6U · 2025-11-25
> >
> > I thank the authors for the thorough rebuttal and the effort in resolving my uncertainties. I think I now have a better understanding, but unfortunately some open questions still remain. I will go through my thoughts on the subquestions below.
> >
> > **g grouping**
> > Thank you for the clarification and confirmation. How do you choose the k value for the top-k largest eigenvalues?
> >
> > **Diffusion model structure**
> > Potentially due to my unfamiliarity with the topic, I still do not entirely understand how the diffusion setup works. Line 212 says that "the spectral representation of a graph signal is $\hat x = U^T x$, where $\hat x$ contains the spectral coefficients." What is $x$ exactly here? The adjacency matrix, or the node feature matrix? The new training loop Algorithm in the Appendix seems to imply that the diffusion is not on the graph eigenvalues, but instead on the entire graph adjacency matrix and the node features (since there is no mention of any eigendecomposition)?
> >
> > One interpretation I can make that the generation is only over the node features and not the adjacency matrix at all? Is this correct?
> >
> > **Role of z**
> > Thank you for the clarification, but I think I still have some confusion here. How is gamma fixed at inference time? What is the point of training with z if z is somehow fixed/not used? What does it mean that $g$ is unchanged during the inference process? Do you use a single $g$ value for generating all samples, or do you give the model the $g$ value extracted for a particular data sample, and that sample is then re-generated? The algorithm seems to imply this. So one way to put it is that the model trained is conditional on some graph properties, and always requires a graph from the training/validation set as input to generate new graphs? Apologies for the flood of questions, but feel to answer with whatever seems to resolve my confusion based on the questions.
> >
> > **Ablations**
> > I thank the authors for the new ablations. Unfortunately, I still do not see why you could not apply z conditioning on the denoiser itself, and not include the adaptive schedule gamma at all. This seems trivial: Simply add it as an input to the neural network in some reasonable some way. It may be that I have still misunderstood how the method works, but as I understand it, this should be doable.
> >
> > On the new ablations, there seems to be an overall pattern where adding the group conditioning does improve validity and avg. MAE over not using z and having a single group, while not degrading the uniqueness and novelty too much. This seems like a useful data point, but I am not sure what is the result that I should make out from this. What is MAE calculating, exactly? Accuracy of property-conditional generation, in the sense that we get a good score if the generated samples have the same properties as we conditioned on? If so, how are the conditions g,z chosen relative to the property condition? I'm worried that there is an information leak in that g,z reveal additional information that is useful for reconstructing the original $X_0$ in the data set.
> >
> > **Figure 3**
> > The plot makes sense to me now! I think it is a relevant plot.
> >
> > **Figure 5**
> > Thank you for the clarification and analysis. I think the overall intuition that graphs with different spectra having different schedules makes sense. It is still a bit odd that, e.g., group 1 graph is below group 2 graph with 3 groups and 7 groups, but not in 5 groups. One would imagine that the pattern would be more monotonic? Do we have some intuition on which noise levels should correspond to which eigenvalue bands? Having something like this would seem to put the model on a stronger grounding in the sense that there is a clear intuition for what we expect learning the schedule to do and help with. But to be clear, this is not my biggest concern at the moment.

---

> ### Author Response · Authors · 2025-12-02
>
> **[R1] $g$ grouping**
> We use a fixed value of k=32, which is admittedly an empirical choice. Through direct observation of the distribution of our datasets, we found that both molecular graphs (QM9, MOSES) and non-molecular graphs (SBM, planar small) typically contain nodes in the range of 20–40. With k=32, we can capture between 70-100% of the total spectrum across most graph datasets, depending on their size.
>
> We determined that increasing k beyond 32 yields diminishing returns in terms of spectral coverage relative to the computational cost, particularly for datasets dominated by small-sized graphs. Importantly, since our work focuses on identifying distributional differences across graph spectra rather than learning the complete spectrum itself, capturing the full spectrum is not strictly necessary for our approach.
>
> While preparing this response, we also found supporting evidence in prior work. In [1], the appendix notes that a number of nodes = 32 provides the best approximation, and [2] similarly observes that performance saturates when using 32 eigenfunctions.
>
> **[R2] Diffusion model structure**
> Thank you for raising this concern. We would like to clarify our methodology, as it appears there may be some confusion. In our framework, the role of Laplacian-based eigenvalues is strictly limited to adapting the noise scheduler to the graph structure.
>
> The actual diffusion process itself remains unchanged from the base model (DiGress or GraphDiT). ADHaG does not implement a separate spectral-domain diffusion. As shown in the training loop in the Appendix I, the diffusion operates on a discrete state space composed of node types and edge types, exactly as in the base models. The forward process, reverse process, transition model, and loss function all follow the spatial-domain categorical diffusion framework used by DiGress/GraphDiT.
>
> In summary, ADHaG does not modify the diffusion mechanism of the base models—it only adjusts the noise scheduler using eigenvalue information.
>
> Therefore, the spectral diffusion description in line 212 should be understood in the conventional sense as referring to graph signals (i.e., node features). We clarify that the graph Fourier transform is used solely to determine which scheduler group each graph is assigned to, and the spectral coefficients $\hat{x}$ are not used in the actual diffusion process.
>
> We apologize if any part of our explanation caused confusion. To aid understanding, we have provided a simple example demonstrating the eigendecomposition process on a sample molecular graph. Additionally, we have added a more detailed pipeline diagram in the Appendix K, which we believe will be helpful in understanding this response.
>
> **[R3] Role of $z$**
>
> **1. How is gamma fixed at inference time?**
> During training, the learned $\gamma_g$ for each group $g$ determines the scheduler shape for that group. At inference time, we use the fixed scheduler corresponding to the group to which the input graph belongs.
>
> **2. What is the point of training with $z$ if $z$ is somehow fixed/not used?**
> $z$ serves as a fundamental feature representing the spectral characteristics of each graph. It enables $\gamma_g$ to be learned in a way that is tailored to the properties of graphs belonging to group g.
>
> **3. What does it mean that $g$ is unchanged during the inference process?**
> As described in the manuscript, $g$ is deterministically assigned to each graph via eigendecomposition prior to training. It acts as a discrete-valued feature assigned to the data, meaning that inference is performed using the scheduler $\gamma_g$ optimized for the spectral characteristics of the given graph.
>
> **4. Do you use a single $g$ value for generating all samples, or do you give the model the $g$ value extracted for a particular data sample, and that sample is then re-generated?**
>
> The former is correct. As explained above, the $g$ value is predetermined through eigendecomposition, independent of any training parameters. Training is then performed conditioned on these pre-assigned $g$ values.
>
> **5. The algorithm seems to imply this. So one way to put it is that the model trained is conditional on some graph properties, and always requires a graph from the training/validation set as input to generate new graphs?**
>
> Our methodology does not differ significantly from conventional diffusion approaches, except that we pre-compute eigenvalues for each graph. Through eigenvalue computation, we extract discrete-valued information $g$ and assign the corresponding scheduler $\gamma_g$ accordingly.
>
> We thank you for your thoughtful engagement with our methodology. If this response requires further clarification, we refer you to the detailed framework diagram newly added to the Appendix K as mentioned above.
>
> > [1] Huang, Yinan, Haoyu Wang, and Pan Li. "What are good positional encodings for directed graphs?." ICLR 2025.
> > [2] Elhag, Ahmed A., et al. "Manifold diffusion fields." ICLR 2024.

---

> ### Author Response · Authors · 2025-12-02
>
> **[R4] Ablations**
> While we could consider directly conditioning $z$ on the denoiser as you suggest, our work is focused on scheduler adaptation. We believe that artificially designing such experiments may not appropriately address your concern, as it shifts the focus from the core premise of our approach.
>
> Regarding your question about MAE: referring to the official GraphDiT GitHub code, MAE measures the difference between the target properties provided as conditions (targets) and the properties of the generated molecules (scores). Therefore, as you correctly understood, it is a metric that evaluates how accurate the property-conditional generation is.
>
> When calculating the properties of generated molecules (sample scores), we do not directly run actual physics simulators (DFT, MD, etc.), but instead use pre-trained surrogate models or RDKit functions as oracles. For more detailed information on the datasets and metrics, please refer to the Appendix of GraphDiT.
>
> Regarding your concern about information leak, as we have explained previously, our method uses the same fundamental diffusion process as the baseline diffusion model. Therefore, compared to the original GraphDiT, there is no additional information beyond the graph spectrum that could contribute to performance improvements.
>
> **[R5] Figure 5**
> In response to your question, we conducted an additional experiment to more clearly demonstrate the benefit of the adaptive scheduler. Specifically, when G=3, we designed an experiment where we used only the scheduler corresponding to g=2 to sample all graphs belonging to groups g=0, 1, 2. This allowed us to examine how performance changes when using a single scheduler regardless of the group assignment.
>
> On the QM9_no_h dataset, we observed an overall decline in performance across uniqueness, novelty, and NLL. Specifically, validity decreased from 99.7% to 99.4%, uniqueness from 95.9% to 95.5%, and novelty from 34.6% to 33.5%. For reference, the baseline DiGress achieves an average novelty of 33.4%. Additionally, NLL increased from 66.8 to 67.6.

---

### Official Review · Reviewer_Tfcb · 2025-10-31

**Soundness:** 2
**Presentation:** 2
**Contribution:** 3
**Rating:** 4
**Confidence:** 4

**Summary:**

This paper proposes ADHaG, an adaptive scheduling framework for graph diffusion models. The key idea is to introduce harmonic grouping, which partitions graphs based on their Laplacian spectra, and to train feature-conditioned schedulers for each group. The approach aims to handle inter-domain diversity (differences across datasets) and intra-domain variability (differences within datasets). Experimental results on several benchmarks (QM9, 10K Polymers, SBM, Planar) show modest improvements over classical baseline diffusion models.

**Strengths:**

- Introducing adaptivity at the scheduler level rather than model architecture is a fresh angle.

- The proposed framework is architecture-agnostic and can, in principle, be integrated into existing diffusion pipelines with minimal implementation overhead.

- Leveraging the Laplacian spectrum to capture both global and local structural variations is in accordance with established principles in graph signal processing.

**Weaknesses:**

1. The paper repeatedly claims to provide a principled and spectrum-aware approach, yet offers no formal justification for why Laplacian-based grouping should lead to improved noise scheduling. The relationship between spectral bins and generative performance is only empirical, not theoretically grounded. No ablation or analytical study explains why spectral similarity should correlate with optimal scheduler parameters.

2. The proposed harmonic grouping trains an independent scheduler for each spectral group. Thus, model size and parameter count increase linearly with the number of groups.
However, the experimental results show no clear correlation between more groups (e.g., G=7) and better performance. The paper fails to quantify this trade-off in terms of memory, runtime, or parameter efficiency.

3. The most recent diffusion-based molecular graph generation model included is DiGress (2023). More recent and relevant baselines should be included.

4. The molecular experiments are restricted to QM9 and 10K Polymers, omitting other widely adopted datasets such as ZINC and MOSES, which are standard in graph generative model evaluation. This limitation weakens the paper’s generality claims for molecular domains.

5. For QM9, the paper only reports Validity and Uniqueness metrics, which are insufficient to evaluate generation quality. Established benchmarks also include Novelty, FCD, NSPDK, and KL-divergence against real distributions.
The current reporting makes the comparison against DiGress, GDSS, and GSDM incomplete and potentially misleading. The authors are strongly encouraged to adopt the same experimental protocols and metric sets as these benchmark models to ensure fairness and comparability.

6. The paper does not mention a code release, nor does it provide sufficient implementation details to ensure reproducibility.

**Questions:**

1. Why have the most widely used benchmarks for molecular graph generation, ZINC and MOSES, been omitted, despite being used by the baselines (e.g., DiGress and GDSS)?

2. The reported results on generic graphs and QM9 differ significantly from the original results in benchmark papers (e.g., GDSS, DiGress) and recent models like GruM (2024). Could the authors clarify whether these differences arise from variations in data preprocessing, evaluation metrics, or training setups?

3. Can the authors provide any analytical or empirical evidence linking eigenvalue distribution to optimal noise scheduling? For instance, does spectral bandwidth correlate with denoising difficulty?

---

> ### Author Response · Authors · 2025-11-22
>
> We appreciate your thoughtful comments and meaningful suggestions for improving our manuscript. We have carefully addressed your key concerns below.
>
> **[W1, Q3] No formal justification & analytical study for why Laplacian-based grouping should lead to improved noise scheduling**
>
> Our use of Laplacian-based grouping is motivated by the spectral bias of neural networks: low-frequency components are learned more easily, whereas high-frequency components demand finer updates [1]. In the graph setting, the Laplacian spectrum orders these structural complexities, with larger eigenvalues corresponding to more irregular modes. Because graphs differ in how much spectral mass they allocate to low- versus high-complexity regions, they induce distinct denoising requirements for the reverse diffusion model.
>
> | Method             | NLL ↓ | Valid ↑ | Unique ↑ |
> |--------------------|-------|---------|-----------|
> | Dataset            | –     | 99.3    | 100       |
> | DiGress            | 69.6  | 99.0    | 96.2      |
> | + Random Group     | 69.8  | 98.8    | 96.6      |
> | + Randequl Group   | 69.9  | 99.2    | 96.4      |
> | + Harmonic Group   | **66.0** | **99.4** | 96.3  |
>
> In the table above, we also provide empirical evidence by replacing the spectrum-based grouping with groups formed entirely at random and training a separate adaptive scheduler for each group. Random Group assigns each graph to a random group, Randequal Group assigns graphs randomly while enforcing equal group sizes, and Harmonic Group uses the proposed Laplacian-based grouping. Because these random partitions (Random, and Randequal) eliminate any relationship between spectral characteristics and grouping, they function as an analytical control. Under this setting, the resulting schedulers fail to improve upon a fixed global schedule and often perform worse, indicating that the gains of our approach emerge specifically when groups capture differences in frequencies rather than from grouping alone. We have added this table to the manuscript as Table 8.
>
>
> [1] Rahaman, Nasim, et al. "On the spectral bias of neural networks." International conference on machine learning. PMLR, 2019.
>
>
> **[W2] Trade-off in terms of memory, runtime, or parameter efficiency**
>
> Thank you for the insightful comment. In theory, constructing the full graph Laplacian incurs up to $\mathcal{O}(V^3)$ complexity (where $V$ and $E$ denote the number of nodes and edges). However, our method does *not* require cubic computation in practice. As noted in Chen et al. [2], the complexity is reduced to $\mathcal{O}(V+E)$ because we restrict the number of eigenvalues to a fixed budget determined by a hyperparameter. As described in Appendix A, we use at most 32 eigenvalues, and this number can be adjusted depending on the task. Moreover, each graph's Laplacian and eigenvalues are computed *once* prior to training and reused throughout the entire learning process, so this step does not slow down training. Regarding memory, our method introduces additional parameters $\pi$ for learning $\gamma$, but the overhead is minimal. These parameters are produced by a lightweight multilayer perceptron and remain substantially smaller than the original diffusion parameters $\theta$. Within the range of groups used in our experiments ($G \in \{3, \dots, 7\}$), the total number of learnable parameters increases only marginally and does not significantly impact memory usage.
>
> [2] Chen, Pin-Yu, et al. "Fast incremental von Neumann graph entropy computation: Theory, algorithm, and applications." International Conference on Machine Learning. PMLR, 2019.
>
> **[W3] More recent and relevant baselines should be included**
>
> We note that GDSS, which is included in our experiment, is an ICLR 2025 paper and therefore represents a recent state-of-the-art baseline. While DiGress (2023) is not as recent, it remains a widely used benchmark model. We believe this combination reflects both the recent and the established baselines in the domain.

---

> ### Author Response · Authors · 2025-11-22
>
> **[W4, Q1] More widely adopted datasets such as ZINC and MOSES**
>
> | MOSES dataset       | Class     | Val ↑ | Unique ↑ | FCD ↓ |
> |--------------|-----------|--------|-----------|--------|
> | VAE          | SMILES    | 97.7   | 99.8      | 0.57   |
> | JT-VAE       | Fragment  | 100    | 100       | 1.00   |
> | GraphINVENT | Autoreg.  | 96.4   | 99.8      | 1.22   |
> | ConGress     | One-shot  | 83.4   | 99.9      | 1.48   |
> | DiGress      | One-shot  | 85.7   | 100       | 1.19   |
> | **ADHaG (ours)** | One-shot  | **86.6** | **100** | **0.84** |
>
> According to the results in the table above, ADHaG outperforms prior one-shot graph generative models on the benchmark. In MOSES, our method achieves higher validity and the best FCD score among one-shot methods, indicating a closer match to the reference distribution.
>
> **[W5] Additional metrics on QM9 (Novelty, FCD, NSPDK, and KL-divergence against real distributions)**
>
> We evaluated the various metrics you suggested on the QM9 experiment. The results are shown in the table below. These results have been added to the Appendix G as Table 11, where the four columns on the right show the results for the added metrics. We found that our methodology outperforms DiGress across all four metrics.
>
> **Ablation study on multi conditional generation according to $z$ conditioning and Harmonic grouping.**
>
>
> | Model | Novelty ↑ | FCD ↓ | NSPDK ↓ | KL ↓ |
> |-------|-----------|-------|---------|------|
> | Dataset | | - | - | - |
> | DiGress (marginal) | 33.4 | 11.6 | 0.28 | 0.23 |
> | + ADHaG (Ours) | **34.0** | **10.7** | **0.27** | **0.22** |
>
>
> **[W6] Code release**
>
> We will release the full code and implementation details upon acceptance to ensure complete reproducibility.
>
> **[Q2] Differing significantly from the original results in benchmark papers (e.g., GDSS, DiGress, and GruM)**
>
> We would like to clarify that all baseline numbers reported in our tables are taken directly from the original benchmark papers, without any reimplementation or modification. Specifically, Table 1 employs the values reported in GraphDiT (Table 1), Tables 2 and 3 draw upon the results from DiGress (Table 2 and Appendix Table 6), and Tables 4 and 5 incorporate the numbers provided in GDSS (Table 1 and Table 5), which were computed by the GDSS authors themselves. For earlier baselines such as GraphRNN and SPECTRE, we follow the citations used in these papers and adopt the exact values they report. Therefore, the differences observed by the reviewer do not stem from variations in our preprocessing, evaluation metrics, or training configurations; we employ the results as stated in the original sources.

---

### Official Review · Reviewer_E8no · 2025-11-01

**Soundness:** 2
**Presentation:** 2
**Contribution:** 2
**Rating:** 2
**Confidence:** 4

**Summary:**

The paper presents a trained adaptive noise scheduling method for denoising diffusion models for graphs (specifically the GDSM variety, i.e. diffusing only the eigenvalues and using training set eigenvectors).

For this, the top k eigenvalues are used to group the graphs into groups via “harmonic similarity” (histogram sorting and binning) and each  is assigned such a group during training and denoising.

the learned schedule is trained as a learning an offset to the scaling exponent of a cosine schedule to ensure it remains within reasonable bands. evaluation is performed on SBM,planar, communitysmall and QM9 datasets , as well as 10k polymers.

**Strengths:**

- originality: using spectral properties to automatically fine tune  the noise schedule is a decently non-obvious idea

- significance: the method  seems to *sometimes* yield strong improvements

- quality: a wide set of baselines and datasets is evaluated, the method is well described

- clarity: the paper is overall clearly written

**Weaknesses:**

- legibility improvement:

    - harmonic similarity is only defined by jointly reading the figure + prose, make clearer how the quantiles are constructed

    - F_phi should be defined in the main body or an explicit reference to appendix should be made

- lines 304 to 312 appear to require further editing, the  “that is… In contrast” construction does not parse to me  (the two paragraphs say the same thing, but are written as contradiction)

- I don’t think from the data presented it can be claimed the method consistently leads to improvements, needs to be evaluated across multiple training seeds and made rigorous e.g. as such:

    - train multiple models

    - check if the framework consistently improves metrics per batch of seeds on a model ( check for statistically significance/compute CI)

    - compute check p(intervention has significant difference) >0.5 => then can make the claim

- Figure 4 requires a baseline comparison without the harmonic grouping (maybe the methods are aleady good at modeling the EV distribution?)

**Questions:**

- the main issue is the statistical significance test from the weaknesses and more careful baselining, showing the actual improvement of the method
- other Question: am I correct that the scheduler is also used to generate noised samples (meaning that the t steps are now variably sized across training, and optimized to minimize loss/finding the easiest path possible?)

- do you  force zero SNR/full noising at noise limit? see [https://openaccess.thecvf.com/content/WACV2024/papers/Lin_Common_Diffusion_Noise_Schedules_and_Sample_Steps_Are_Flawed_WACV_2024_paper.pdf](https://openaccess.thecvf.com/content/WACV2024/papers/Lin_Common_Diffusion_Noise_Schedules_and_Sample_Steps_Are_Flawed_WACV_2024_paper.pdf)

---

> ### Author Response · Authors · 2025-11-22
>
> We appreciate your thoughtful comments and meaningful suggestions for improving our manuscript. We have carefully addressed your key concerns below.
>
> **[W1] Legibility improvement (harmonic similarity, and $\mathcal{F}_{\phi}$)**
>
> First, we apologize for causing confusion by not clearly defining the terminology. Harmonic similarity refers to the explanation provided in lines 254-258 of Section 4.1, where when dividing the entire eigenvalue range into G equal-width groups, two graphs classified into the same group are graphs with relatively similar dominant eigenvalue values, and we expressed this as having high *'harmonic similarity'* in Figure 1. We have clarified the caption of Figure 1 more specifically. And, $\mathcal{F}_{\phi}$ plays a role in learning raw values of a scalar $\gamma$ of group $g$ at time $t$. Through normalization via integration and min-max range adjustment, the final scheduler $\gamma$ that operates in a numerically stable manner is obtained.
>
> **[W2] Written as contradiction (lines 304 to 312)**
>
> Sorry for the confusion caused by the original description. We have revised lines 304–312 of the main manuscript to eliminate the contradictory phrasing and ensure the explanation reads clearly.
>
> **[W3, Q1] Evaluation across multiple training seeds**
>
> We apologize for the confusion. All reported results, except for the SBM and Community datasets, represent averages over three independent runs with different seeds. To clarify this and avoid ambiguity regarding statistical significance and baseline comparisons, we have stated this more explicitly in the revised version (line 365 in Section 5.1).
>
> **[W4] Figure 4 requires a baseline comparison without the harmonic grouping**
>
> To more clearly demonstrate the effectiveness of harmonic grouping, we performed ablation studies on two datasets, 10K Polymers and QM9, with results presented in newly added Tables 7 and 8, respectively. Table 7 shows that without grouping (G=1), both validity and Avg. MAE decreases compared to when grouping is applied. However, the effects of grouping were not observed in Uniqueness and Novelty. This trend was also evident in QM9, where using criteria other than eigenvalue similarity (Random, Randequal) resulted in decreased NLL and Validity, but showed no significant difference in Uniqueness.
> In summary, grouping demonstrates effectiveness in improving validity and reducing error, but does not show significant effects on novelty and uniqueness. Since these are empirical observations, a precise investigation into the underlying causes is somewhat limited at this stage.
>
> **[Q2] Are $t$ steps variably sized across training?**
>
> Thank you for the clarifying question. If we understand your concern correctly, our method does not optimize the timestep itself; rather, it learns the noise magnitude assigned to each fixed timestep. The number and spacing of timesteps remain unchanged throughout training. If our interpretation is inaccurate, please feel free to let us know, and we will clarify accordingly.
>
>
> **[Q3] Forcing zero SNR/full noise at noise limit?**
>
> Thank you for the helpful suggestion. We use the cosine scheduler without clipping, which—as noted in the suggested paper—already decays sufficiently close to zero and therefore does not require an additional “force-zero” mechanism. In our implementation, the terminal noise level reaches approximately 1 / 100,000, which is two to one hundred times smaller than the bias values reported in that work. Accordingly, we believe that the process effectively achieves full noising at the noise limit.

---

> > ### Comment · Reviewer_E8no · 2025-11-25
> > **Thank for your rebutall**
> >
> > W1: thank you
> > W2: thank you
> > W3/Q1: to clarify, are these independent _training_+sampling runs or independent sampling runs only?
> > W4: Thank you, while I think the ablations are independnetly useful,they still miss the point of my remark a bit, I would like to see how well the baselines match the eigenvalue distributions (e.g. measured as 1D wasserstein distance between the histograms as well as simply visualized) to check whether the spectral matching is meaningful.
> > Q2: Sorry, yes, that got at my intent despite being weirdly phrased.  So basically, the training in theory allows the network to reduce the total noise applied (making the denoising easier? This linksinto my Q3, but based on your answer, you would think that the noise is sufficiently to reach a full noise limit (as is also borne out by numerics I guess)?

---

> ### Author Response · Authors · 2025-12-02
>
> **[W3/Q1] Independent training + sampling runs vs. sampling-only runs**
>
> The experimental results are obtained by averaging over three independent training and sampling runs. For each run, we reinitialize the random seed and retrain both the diffusion model parameters 𝜃 and the ADHaG scheduler parameters 𝜙 from scratch, and then generate a new set of samples for evaluation.
> Thus, the reported numbers reflect the mean over three independent models, not multiple sampling from a single trained model.
>
>
> **[W4] Spectral matching: eigenvalue distribution comparison**
>
> Following your suggestion, we computed the 1D Wasserstein distance between the eigenvalue histograms of the generated graphs and those of the test set. Across datasets, ADHaG consistently achieves lower distances (10–20% relative improvement) than DiGress, indicating more accurate spectral alignment.
>
> **QM9-H:**
> - ADHaG: 0.002813
> - DiGress: 0.003281
>
> **10K Polymers:**
> - ADHaG: 0.004056
> - DiGress: 0.005287
>
> For the polymer dataset, the distribution visualizations are already provided in the main manuscript (Figure 4), and they likewise show that harmonic grouping yields noticeably better alignment with the ground-truth eigenvalue distribution. We also added the corresponding eigenvalue histogram visualizations for the polymer dataset in Appendix J.
>
>
> **[Q2, Q3] Does γ reduce the total noise? Does ADHaG preserve full-noise behavior?**
>
> In ADHaG, the learned $\gamma(t,z)$ does not modify the timestep schedule nor reduce the overall amount of noise applied during the forward process. Instead, $\gamma$ rescales the base scheduler $\bar{\alpha}(t)$ in an exponential form, reshaping the profile of $\log \bar{\alpha}(t)$ across groups rather than altering its magnitude. Importantly, $\gamma$ converges to values around $2.0$ at the final timestep, meaning that if the base scheduler reaches a full-noise limit, the ADHaG-modified scheduler does so as well. Thus, ADHaG does not interfere with or weaken the full-noise property of the underlying diffusion process, nor does it reduce the total noise applied.
>
> In summary, ADHaG preserves the full-noise limit of the base scheduler; it only adjusts the distributional shape of the schedule across groups.

---

### Official Review · Reviewer_mnTU · 2025-11-01

**Soundness:** 3
**Presentation:** 3
**Contribution:** 2
**Rating:** 6
**Confidence:** 4

**Summary:**

The authors proposes a novel adaptive noise scheduler for graph generative models based on diffusion. This is certainly one important factor and design aspect that could help improve graph generation performance. The authors propose to group graphs according to spectral properties, and then learns a feature-conditioned scheduler for each group. The authors show performance that are competitive and slightly better on some perspective, with respect to recent graph generative models on QM9 and synthetic graph datasets.

**Strengths:**

Improving the noise scheduler is certainly an interesting direction for developing better graph generation models. The idea of noise scheduler that depends on graph properties is natural but a good idea. And developing different schedulers for each group of graphs is reasonable and novel. The performance of the proposed method on different datasets seems to confirm the benefits of the proposed graph scheduler.

**Weaknesses:**

Grouping graphs is certainly a good idea for designing a finite set of noise schedulers. It is however not fully clear that inter-diversity and intra-variability of graphs have to be considered on the same level: it seems that large graph variations across data/applications is handled separately at the end, even if this is not fully clear in the first part of the paper. Also, one might imagine that large variations across datasets can be handled differently than training a fully universal adaptive scheduler (e.g., by considering different generative models for graph datasets that are completely different).

The choice of grouping graphs by their spectral properties looks natural. However, it maybe lacks a strong convincing argument that this is the best way to adapt noise schedulers. Wouldn't it depend on the actual diffusion model? It seems to be mostly applied in combination with DiGress: even there, what is the exact connection between noise scheduling and spectral properties of graphs? (Btw, DiGress does not look like a spectral diffusion model, despite what is written in the present paper).

The conditioning block 'z' is described very shortly in the main paper, it is difficult to really appreciate its construction and benefits.

The design of the adaptive noise scheduler makes sense. However, the proposed idea can probably be applied to other parametric forms than (3) - which is probably not a unique solution. It may be good to motive this particular form, or discuss alternative forms, or at least clarify that the solution is not unique, and maybe not optimal (there is no claim of optimality in the paper ofc, but it may still be good to clarify things).

Experiments are usually well chosen, and illustrate the benefits proposed by the adaptive noise scheduler. However, QM9 is a very small dataset, and state-of-the-art generative models like those based on flow matching, are missing. This is not a critical issue, as again the authors do not claim any optimal generation performance. Yet, the text can be clarified in that respect, and the choice of experiments can be motivated accordingly.

From Table 5, it seems that most gains come from validity improvement. Whose value stays relatively low. It may be good to explain why benefits are essentially observed in terms of validity, and what further bottlenecks could lead to even better validity scores.

Computational complexity is not discussed. In general, the diffusion process should not be more complex than competitors. Yet, two new components may deserve a short discussion: the construction of $\gamma_g$ and the spectral grouping (which apparently relies in eigendecomposition of graphs).

**Questions:**

see above

---

> ### Author Response · Authors · 2025-11-22
>
> We appreciate your thoughtful comments and meaningful suggestions for improving our manuscript. We have carefully addressed your key concerns below.
>
>
> **[W1]  Inter-diversity and intra-variability of graphs on the same level**
>
> Sorry for not explaining this distinction clearly. Our intention was not to imply that inter-diversity and intra-variability should be addressed at the same level. Instead, our point was to underscore that, relative to image domains commonly considered in diffusion-based research, graph domains possess intrinsically more complex structural properties—such as a markedly more intricate underlying manifold—which render the learning process substantially more demanding. Unlike images, graphs exhibit substantial heterogeneity both across datasets—for example, social networks versus molecular graphs differ fundamentally in their node and edge semantics—and within a single dataset, where graph sizes and structural patterns can vary significantly. We use the terms inter-diversity and intra-variability to distinguish these two sources of heterogeneity, and this distinction is illustrated in Figure 1 in the main manuscript.
>
> **[W2] Does the choice of grouping depend on the diffusion model?**
>
> We would like to clarify that our method is not designed specifically for DiGress. To assess its generality, we additionally applied the proposed adaptive scheduling strategy to GraphDiT, which differs substantially from DiGress in both architecture and diffusion formulation. In both models, we observed consistent improvements, indicating that the effectiveness of our spectral grouping is not tied to a particular diffusion framework but extends across distinct graph diffusion paradigms. To further confirm the exact connection between noise scheduling and spectral properties of graphs, we compare our spectrum-based grouping strategy against a no-grouping baseline (i.e., G=1) and observe that grouping graphs according to their spectral similarity yields consistently superior performance.
>
> | Metric                               | Graph-DiT + ADHaG        | Graph-DiT + No Grouping (G=1) |
> |--------------------------------------|--------------|--------------------|
> | FCD_structural (↓ better)            | **0.854918** | 1.155978           |
> | NSPDK_MMD (↓ better)                 | **0.961187** | 0.988782           |
> | NSPDK_MMD2 (↓ better)                | **0.923879** | 0.977690           |
> | KL_ref_gen (↓ better)                | **0.002018** | 0.005983           |
> | KL_gen_ref (↓ better)                | **0.001907** | 0.006254           |
> | KL_symmetric (↓ better)              | **0.001963** | 0.006119           |
>
> **[W3] Construction and benefits of ‘$z$’**
>
> $z$ serves as a conditioning signal. It is obtained by embedding the input graph $x$ through a 2-layer graph isomorphism network (GIN), which allows the adaptive noise scheduler $\gamma$ to incorporate additional local structural information from the graph. This mechanism is not specific to GIN; any GNN architecture may be substituted without loss of generality. We have clarified the role of the conditioning variable $z$ in Section 4.1 of the manuscript and have added corresponding pseudocode in Appendix I.
>
> **[W4] Other parametric forms than Eq. (3)**
>
> Thanks for pointing this out. We do not claim that the parametric form in Eq. (3) is unique. Following the normalized formulation used in MuLAN [1], we scale the adaptive $\gamma$ to ensure stable and efficient convergence. We adopt the cosine scheduler as the base form, and the $\gamma$ values are independently learned for each spectrum group, as described in Appendix C. While alternative parametric forms are certainly possible, our design provides a stable and effective way to incorporate graph properties into noise scheduling.
>
> [1] Sahoo, Subham, et al. "Diffusion models with learned adaptive noise." Advances in Neural Information Processing Systems 37 (2024): 105730-105779.

---

> ### Author Response · Authors · 2025-11-22
>
> **[W5] Experiments on a large dataset and flow-matching models**
>
> Thank you for the suggestion. We agree that demonstrating the effectiveness of our method on larger datasets is an important point. Following this, we additionally evaluate our approach on the MOSES dataset [2] (approximately 2 million molecules) and the GuacaMol dataset [3] (approximately 1.3 million molecules), both of which are widely used in drug discovery. Notably, GuacaMol includes molecules with up to 100 atoms and a much broader set of atom types compared to QM9. We conducted full training and evaluation on these datasets and reported its results in the table below and Table 9,10 in Appendix G.
>
>
> | MOSES dataset       | Class     | Val ↑ | Unique ↑ | FCD ↓ |
> |--------------|-----------|--------|-----------|--------|
> | VAE          | SMILES    | 97.7   | 99.8      | 0.57   |
> | JT-VAE       | Fragment  | 100    | 100       | 1.00   |
> | GraphINVENT | Autoreg.  | 96.4   | 99.8      | 1.22   |
> | ConGress     | One-shot  | 83.4   | 99.9      | 1.48   |
> | DiGress      | One-shot  | 85.7   | 100       | 1.19   |
> | **ADHaG (ours)** | One-shot  | **86.6** | **100** | **0.84** |
>
> | GuacaMol dataset         | Class     | Valid ↑ | Unique ↑ | KL div ↑ | FCD ↑ |
> |----------------|-----------|---------|-----------|-----------|--------|
> | LSTM           | Smiles    | 95.9    | 100       | 99.1      | 91.3   |
> | NAGVAE         | One-shot  | 92.9    | 95.5      | 38.4      | 0.9    |
> | MCTS           | One-shot  | 100     | 100       | 82.2      | 1.5    |
> | ConGress       | One-shot  | 0.1     | 100       | 36.1      | 0.0    |
> | DiGress        | One-shot  | 85.2    | 100       | 92.9      | 68.0   |
> | **ADHaG (ours)** | One-shot  | **88.8** | **100** | **92.9**  | **68.4** |
>
>
> According to the results in the tables above, ADHaG consistently outperforms prior one-shot graph generative models on both benchmarks. In MOSES, our method achieves higher validity and the best FCD score among one-shot methods, indicating a closer match to the reference distribution. In GuacaMol, ADHaG attains competitive or superior performance across all metrics, including improved validity and perfect uniqueness.
>
>
> Also, flow matching does not involve a noise–denoise process or any discrete denoising dynamics; instead, it learns a continuous vector field for ODE-based transport. Because our contribution focuses on adaptive noise scheduling within diffusion-based denoising trajectories, it cannot be applied directly to flow matching, which lacks the corresponding noise-level parameterization.
>
> [2] Polykovskiy, Daniil, et al. "Molecular sets (MOSES): a benchmarking platform for molecular generation models." Frontiers in pharmacology 11 (2020): 565644.
>
> [3] Brown, Nathan, et al. "GuacaMol: benchmarking models for de novo molecular design." Journal of chemical information and modeling 59.3 (2019): 1096-1108.

---

> ### Author Response · Authors · 2025-11-22
>
> **[W6] Huge gain on validity**
>
> Thank you for your insightful observation. We were also aware of the phenomenon where improvement occurs predominantly in validity performance. In this regard, we compared two experimental results of ADHaG (200 epochs with G=3) and ADHaG (1000 epochs with G=3) in Table 2 (QM9 generation) of our current manuscript: as the number of epochs increases, validity improves but uniqueness somewhat decreases. We considered that these conflicting results indicate that the factors driving these two metrics are different. Given existing research findings that graph kernels in GNNs primarily learn low-frequency regions, we hypothesized that as training progresses (or with more training), generated graphs would emphasize low-frequency regions.
> Based on this, we formulated a hypothesis that validity is more closely related to low-frequency characteristics, while uniqueness is more associated with high-frequency characteristics. When we trained on another dataset, Planar, evaluated after 20,000 and 100,000 epochs, and plotted the eigenvalue distributions of the generated samples (we added Figure 8 and Table 12 in Appendix H), we found that at 20,000 epochs (the relatively shorter training period), more samples with higher eigenvalues were generated, whereas at 100,000 epochs, while the distribution became more similar to that of the original samples, the proportion of samples with higher eigenvalue distributions relatively decreased.
>
> This suggests that during the training process, when improving the quality of generated molecules, molecules in the higher eigenvalue region are being disproportionately filtered out, though at this stage it is difficult to clearly explain why this phenomenon occurs. Although this trade-off between validity and uniqueness was discovered empirically, it was also difficult to find existing theoretical grounds that could directly support it. These analytical results suggest that there may be potential limitations to achieving uniform performance improvements in molecule generation using diffusion. We believe this is a research topic we can explore in depth in future work.
>
> **[W7] Computational complexity**
>
> Thank you for the insightful comment. In theory, constructing the full graph Laplacian incurs up to $\mathcal{O}(V^3)$ complexity (where $V$ and $E$ denote the number of nodes and edges). However, our method does \emph{not} require cubic computation in practice. As noted in Chen et al. [3], the complexity is reduced to $\mathcal{O}(V+E)$ because we restrict the number of eigenvalues to a fixed budget determined by a hyperparameter. As described in Appendix A, we use at most 32 eigenvalues, and this number can be adjusted depending on the task. Moreover, each graph's Laplacian and eigenvalues are computed \emph{once} prior to training and reused throughout the entire learning process, so this step does not slow down training. Regarding memory, our method introduces additional parameters $\pi$ for learning $\gamma$, but the overhead is minimal. These parameters are produced by a lightweight multilayer perceptron and remain substantially smaller than the original diffusion parameters $\theta$. Within the range of groups used in our experiments ($G \in \{3, \dots, 7\}$), the total number of learnable parameters increases only marginally and does not significantly impact memory usage.
>
> [4] Chen, Pin-Yu, et al. "Fast incremental von neumann graph entropy computation: Theory, algorithm, and applications." International Conference on Machine Learning. PMLR, 2019.

---

### Author Response · Authors · 2025-12-03
**General Response (cont'd)**

### **Major revisions and additions in the revised manuscript**

Guided by the reviews and subsequent discussion phase, we substantially revised and extended the paper.

First, we clarified the distinction between inter-diversity (across datasets) and intra-variability (within a dataset), rewrote the previously confusing paragraph, and refined the description of harmonic grouping procedure, providing a clearer step-by-step explanation in Appendix I. We also made explicit that Laplacian eigenvalues are used only to adapt the scheduler, while the diffusion process itself remains that of the base models (DiGress / GraphDiT) on discrete node/edge types, and we added a pipeline-style illustration linking eigenvalue preprocessing, ADHaG, and the base diffusion model in Appendix K.

To better justify our design, we expanded the analysis of grouping by adding ablations that compare harmonic grouping with no grouping (G=1) and with random/randequal partitions, showing that only Laplacian-based grouping consistently improves NLL and validity (summarized in Appendix F); we connected this behavior to spectral bias and the role of per-band schedulers for high-eigenvalue regimes.

On the experimental side, we extended molecular experiments to MOSES and GuacaMol, where ADHaG achieves competitive or superior performance (including the best FCD among one-shot methods on MOSES). For QM9, we added additional metrics—Novelty, FCD, NSPDK-MMD, and KL divergence—showing that ADHaG improves or matches DiGress across all of them, summarized in Appendix G, and we included 1D Wasserstein comparisons between eigenvalue histograms of generated and test graphs, where ADHaG consistently achieves lower distances in Appendix J.

We also clarified that, except for SBM and Community, all reported results in the manuscript are averages over three independent training + sampling runs with reinitialized models and schedulers, rather than multiple samplings from a single model.
And then, we discussed computational aspects, noting that Laplacian computation is performed once per graph with truncation to a small number of eigenvalues, that the per-group $\gamma$-MLPs add only a small parameter and memory overhead relative to the base diffusion model, and that our spectral budget is supported by dataset statistics and prior work.

We further clarified the roles of $z$, $g$, and $\gamma$: $g$ is a deterministic, pre-computed group index from eigenvalues and fixed per graph, while $z$ is a GNN-based embedding used only during training to learn $\gamma$; at inference, each group uses a fixed scheduler depending only on $g$.

Multi-conditional ablations on the Polymer O2-N2-CO2 task show that spectral grouping and $z$-conditioning improve validity and property-MAE while keeping uniqueness and novelty competitive and are reported in Appendix F.

We also analyzed the learned γ, and observed that, when applied as an exponential scaling of the base cosine schedule, the resulting schedule still yields a terminal SNR that is effectively zero. Thus ADHaG maintains the same full-noise terminal condition as the base schedule; γ reshapes how noise is allocated across timesteps and spectrum groups (e.g., relatively attenuating some regions while amplifying others) without decreasing the total noise level over the diffusion trajectory.

To support reproducibility, we committed to releasing full code and implementation details upon acceptance.


--------
We hope that these revisions and additional experiments address the reviewers’ main concerns regarding clarity, justification of spectral grouping, experimental robustness, and empirical scope. We are grateful again for the reviewers’ insightful feedback, which substantially improved the quality and transparency of the paper.

---

### Author Response · Authors · 2025-12-03
**General Response**

We sincerely thank all reviewers (mnTU, E8no, Tfcb, and Uz6U) for the time and care invested in evaluating our work and for the many constructive suggestions that helped us improve the paper and clarify our methodology. Before addressing individual comments (in the rebuttal and the revised manuscript), we summarize below how the reviews characterize the strengths of ADHaG and what major changes we made in response.

### **Strengths of ADHaG highlighted by the reviewers**

The reviews highlighted the following key strengths of ADHaG. First, regarding motivation and idea, reviewers noted that adapting the noise scheduler—rather than the model architecture—is a natural yet under-explored way to improve graph diffusion, and that using spectral properties to define data-dependent schedulers is “decently non-obvious” and well motivated.

Second, in terms of harmonic grouping and spectrum-aware design, the idea of grouping graphs by Laplacian spectra and learning separate schedulers for each group was considered reasonable, novel, and aligned with graph signal-processing principles, such as ordering modes from low- to high-frequency.

Third, as an architecture-agnostic framework, reviewers appreciated that ADHaG is compatible with multiple base models (DiGress and GraphDiT in our experiments) and can be integrated into existing pipelines with modest implementation overhead.

Finally, in terms of empirical results, the reviewers recognized that our method often achieves meaningful improvements over strong diffusion baselines on both synthetic and molecular datasets, and that the experimental setup is broad in terms of datasets and baselines.

---

### Meta-Review · Area_Chair_bB1w · 2025-12-07

**Summary:**

This paper introduces ADHaG, an adaptive noise‑scheduling framework for graph diffusion models. The method first groups graphs based on their Laplacian spectral statistics (“harmonic grouping”), and then learns group‑specific schedule modulations—conditioned on structural features during training—that can be seamlessly incorporated into existing diffusion pipelines.

**Reviewer feedback (pre-rebuttal).** Reviews were mixed: reviewers generally liked the direction (scheduler adaptivity as an under-explored lever; spectrum-based grouping; architecture-agnostic integration), but raised concerns about (a) clarity and potential confusion about whether diffusion happens in the spectral domain and how z/grouping is used at inference, (b) limited justification that Laplacian grouping is the “right” choice (mostly empirical), (c) insufficient rigor around robustness/statistical significance and missing ablations (no grouping / random grouping), and (d) experimental scope (datasets/metrics) and practical overhead discussion.

**After rebuttal.** The authors addressed several of the concrete issues raised—particularly ambiguous descriptions and missing experimental controls. The revised manuscript clarifies the role of eigenvalues (scheduler adaptation only), explains the use of $z$, adds multi‑seed runs, and includes ablations comparing harmonic grouping to random partitions and no grouping. Additional metrics and datasets improve the empirical scope.

However, I share the deeper concerns raised by multiple reviewers that the core justification for the proposed method remains largely intuitive and empirical. The rebuttal expands on empirical phenomena (e.g., spectral bias, differences in eigenvalue distributions, improved spectral alignment), but these observations do not yet establish why Laplacian-based grouping should be expected to yield systematically better schedules, nor do they convincingly rule out that the observed gains arise from implementation‑level effects rather than a principled exploitation of graph structure.

Furthermore, despite added explanations, some parts of the presentation and terminology remain difficult to follow, and several passages rely on informal or overloaded language. A more disciplined exposition—precise definitions, cleaner separation of concepts, and a more explicit statement of hypotheses and limitations—would substantially strengthen the work.

For these reasons, while the paper presents an interesting direction, I believe it requires further refinement and investigation before being ready for acceptance.

**Reviewer Concerns:**

The authors have made several improvements in the revision, and some of the reviewers’ concerns were directly addressed:

**Clarification of technical ambiguities:**
The authors resolved a number of points of confusion in the original submission, particularly regarding how eigenvalues, grouping, and conditioning variables are used in the method. These clarifications improve the conceptual correctness of the paper.

**Expanded empirical evidence:**
The authors added new experiments—including additional datasets, metrics, baselines, and ablations—to strengthen the empirical foundation supporting the effectiveness of the approach.

Still outstanding / only partially addressed:

**Mechanism behind the method’s effectiveness:**
The key conceptual concern remains largely unresolved. While the empirical results are now more complete, the paper still does not provide a convincing explanation—analytical or conceptual—for why Laplacian-based grouping should lead to improved scheduling, or whether the gains stem from fundamental modeling principles or from empirical/engineering artifacts. This remains the main gap in the scientific contribution.

**Presentation and precision of exposition:**
Although improved, the presentation still contains places where notation, terminology, and conceptual roles remain difficult to follow. A more disciplined and concise exposition would help make the method easier to understand and evaluate.

**Reviewer Scores:**

- Reviewer mnTU (Originally 6) Given the added clarifications and expanded experiments/complexity discussion, I expect mnTU would remain at the original positive score 6.

- Reviewer E8no (Originally 2) largely on missing multi-seed rigor and missing baselines/spectral-match checks.  With the new multi-run training protocol clarification and Wasserstein spectral matching, plus no-grouping/random-grouping ablations, I expect an increase to 4 (borderline).


- Reviewer Tfcb (Originally 4): Added MOSES/metrics/code-release and compute discussion likely resolves most practical concerns; However, the main concern about the formal justification remain. Therefore, I expect it stay at 4

- Reviewer Uz6U (Originally 2) due to major clarity concerns and requested ablations.  While clarity about inference and the diffusion mechanism is substantially improved, the “denoiser conditioning only” ablation concern is still not fully met. I expect a modest increase to 4.

---

### Decision · Program_Chairs · 2026-01-26

Reject